# SMALL VECTORS, BIG EFFECTS: A MECHANISTIC STUDY OF RL-INDUCED REASONING VIA STEERING VECTORS

## ABSTRACT

The mechanisms by which reasoning training reshapes LLMs' internal computations remain unclear. We study lightweight steering vectors inserted into the base model's residual stream and trained with a reinforcement-learning objective. These vectors match full fine-tuning performance while preserving the interpretability of small, additive interventions. Using logit-lens readouts and path-patching analyses on two models, we find that (i) the last-layer steering vector acts like a token-substitution bias concentrated on the first generated token, consistently boosting tokens such as "To" and "Step"; (ii) the penultimate-layer vector leaves attention patterns largely intact and instead operates through the MLP and unembedding, preferentially up-weighting process words and structure symbols; and (iii) middle layers de-emphasize non-English tokens. Next, we show that a SAE isolates features associated with correct generations. We also show that steering vectors (i) transfer to other models, (ii) combine across layers when trained in isolation, and (iii) concentrate magnitude on meaningful prompt segments under adaptive token-wise scaling. Taken together, these results deepen understanding of how trained steering vectors shape computation and should inform future work in activation engineering and the study of reasoning models.

## 1 INTRODUCTION

Reasoning-oriented language models have recently made striking gains Jaech et al. (2024); Guo et al. (2025). Many top systems are trained with reinforcement learning on verifiable tasks, especially mathematics, where correctness provides reliable rewards. Yet we still lack a mechanistic account of what this training changes inside the network.

We train *steering vectors* – learned additive directions injected into the residual stream, while freezing the base model. This parameterization was shown to match the performance of fully-tuned models (Sinii et al., 2025) and it isolates a small set of features that can be probed, ablated, and composed with mechanistic-interpretability tools. To isolate layer-wise effects, we fit one steering vector per layer $\ell$ and measure its behavioral impact, then analyze in depth the layers with the clearest effects. Our findings are:

- **Layer-wise isolation of RL-induced gains.** We insert a steering vector at a single layer and report the performance achievable by this minimal intervention, averaged across six modern math benchmarks.
- **Two mechanisms for single-layer steering.** Single-layer steering operates via two distinct mechanisms: all pre-final layers downweight non-English tokens, while the final layer modifies the first generation token.
- **Last layer behaves like first-token substitution.** The final-layer vector acts at unembedding, boosting opening tokens (e.g., "To"/"Step"); simply prefixing that token recovers $\sim$10–11 points – about three-quarters of the explicit last-layer gain.
- **Penultimate-layer vector acts through the MLP.** The effect is mediated almost entirely by the MLP, with minimal reliance on attention.
- **Properties of steering vectors.** They compose across layers, transfer to other models, and, with adaptive magnitude, fire selectively on specific tokens.

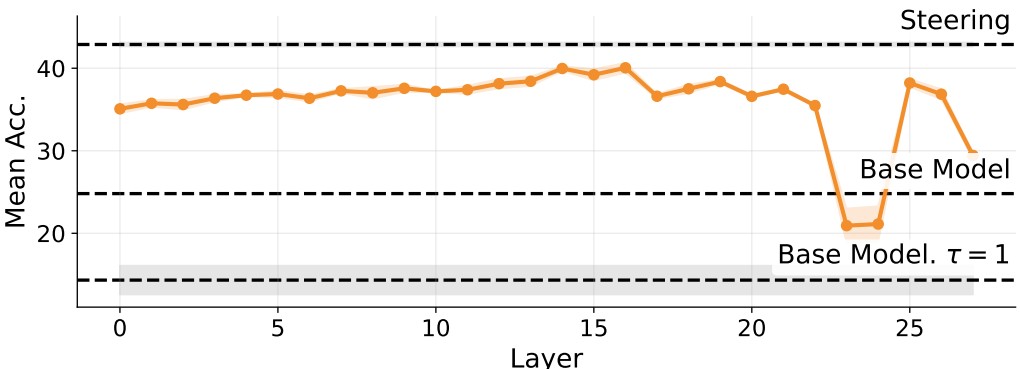

Figure 1: **Single-layer steering.** Mean accuracy on six benchmarks for Qwen2.5-Math-7B when training a single vector $s_\ell$ at layer $\ell$ with all other layers frozen. Mid-layer vectors yield the largest gains but never match all-layer steering, indicating the improvement is distributed across layers.

## 2 RELATED WORK

**Reinforcement learning with verifiable rewards.** Jaech et al. (2024) demonstrated the striking performance of RL-tuned reasoning models, sparking a wave of follow-ups that develop these models (Guo et al., 2025; Zeng et al., 2025; Liu et al., 2025a; Hu et al., 2025). Subsequent work has examined why this training is effective, analysing model behaviour and the sources of its gains (Wang et al., 2025; Ye et al., 2025; Shao et al., 2025; Liu et al., 2025b). We contribute with a mechanistic study of the changes induced by reasoning training.

**Steering vectors** are small additive perturbations to the residual stream that modulate model behavior. They are widely viewed as *feature amplifiers* – strengthening existing computations rather than introducing new mechanisms – and have been used to toggle or amplify *reasoning-like* behaviors (Venhoff et al., 2025; Ward et al., 2025). A common way to obtain them is *contrastive extraction* from activation pairs (e.g., positive vs. negative sentiment) (Turner et al., 2023; Panickssery et al., 2023; Liu et al., 2023; Zou et al., 2023). Beyond extraction, steering directions can also be *trained*: optimized with preference data for controllable generation (Cao et al., 2024), or learned as simple additive vectors that surface latent behaviors such as step-by-step reasoning or self-reflection (Mack & Turner, 2024; Engels et al., 2025; Betley et al., 2025).

In this work, we interpret steering vectors trained with GRPO-like objective using standard tools from mechanistic interpretability – `logit-lens` to read out token-level effects (nostalgebraist, 2020), *path patching* to localize circuits (Wang et al., 2022), and circuit-style analyses in the QK/OV framework (Elhage et al., 2021).

## 3 BACKGROUND

Recent work has shown that training lightweight steering vectors can match the performance of fully trained models (Sinii et al., 2025). Concretely, a vector $s_\ell \in \mathbb{R}^d$ is added to the output residual stream of $\ell$'th layer, and all other weights remain fixed. They used RLOO (Ahmadian et al., 2024) objective to train reasoning models

$$\nabla_\theta J = \mathbb{E}_{x\sim\mathcal{D},\, y\sim\pi_\theta(\cdot|x)}\big[\, a(x,y)\, \nabla_\theta \log \pi_\theta(y \mid x)\,\big],$$

where the advantage is defined as

$$a(x,y) = r(x,y) - b(x), \qquad b(x) = \frac{1}{N}\sum_y r(x,y).$$

Here $r(x,y)$ is the scalar reward for completion $y$ on prompt $x$, and $b(x)$ is the per-prompt baseline for variance reduction.

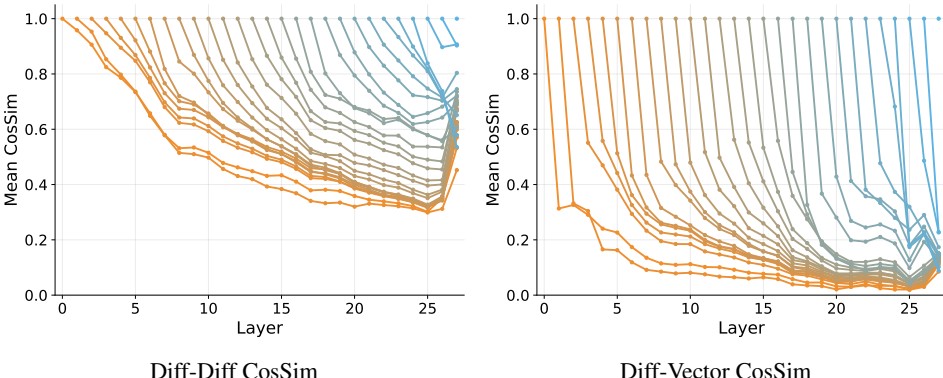

Diff-Diff CosSim                Diff-Vector CosSim

Figure 2: **Steering Vector Persistence.** For each steering layer $i$ (color encodes $i$; warm = early, cool = late) and each target layer $\ell$ on the $x$-axis, we compute the mean cosine similarity of the per-token change in hidden representations $\Delta F_{<\ell,i}$. *Left:* similarity between $\Delta F_{<l,i}(x)$ and the dataset mean $\mathbb{E}_x[\Delta F_{<l,i}(x)]$, showing how aligned the per-token shifts are. *Right:* similarity between $\Delta F_{<l,i}(x)$ and the layer-$\ell$ steering vector $s_\ell$, showing the alignment of the shifts with the layer's own steering vector.

Sinii et al. (2025) argue that this parameterization localizes training-induced changes in the model's internal computations, making the intervention easier to interpret. We adopt this setup to learn per-layer steering vectors for our interpretability study.

## 4 SINGLE-LAYER STEERING VECTORS

**Setup.** We study two base models – Qwen2.5-Math-7B (Team, 2024) and Llama3.1-8B-Instruct (Grattafiori et al., 2024). Models are trained on the Deep-ScaleR dataset (Luo et al., 2025) with the sampling temperature $\tau = 1.0$, a 4K context window for Qwen2.5-Math-7B, and 8K for Llama3.1-8B-Instruct. Rewards are assigned with `Math-Verify`[1]. We used 128 prompts and 16 generations per gradient step. Evaluation spans six math benchmarks: AIME24/25, AMC23, MATH500 (Hendrycks et al., 2021), MinervaMath (Lewkowycz et al., 2022), and Olympiad-Bench (He et al., 2024). We report the mean score across these benchmarks. For MATH500, MinervaMath, and Olympiad-Bench we report PASS@1; for AIME24/25 and AMC23 we report AVG@32 due to their smaller sizes. During evaluation, models decode with sampling at $\tau = 1.0$ following Zeng et al. (2025). Evaluation context length is 4K and 32K for Qwen2.5-Math-7B and other models respectively. All metrics are averaged over three evaluation seeds.

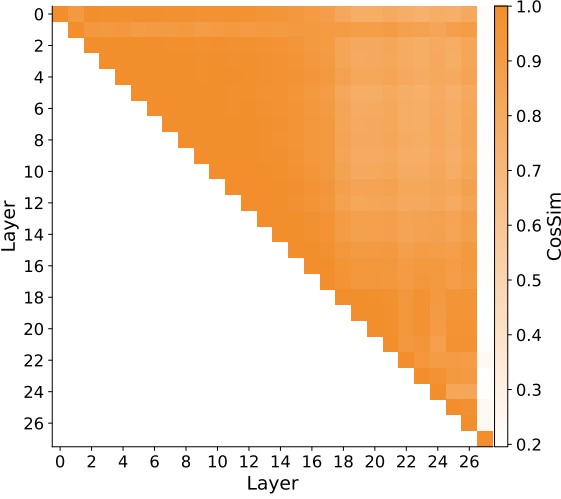

Figure 3: **Similarity of steering-induced unembedding biases.** Each cell shows the cosine similarity between the average final-layer shifts $\mathbb{E}[\Delta F_{<L,i}]$ and $\mathbb{E}[\Delta F_{<L,j}]$ induced by steering at layers $i$ and $j$. High similarity across $i, j < L$ indicates a shared effect on the unembedding regardless of where steering is applied. The last-layer shift implements another mechanism.

---

[1]https://github.com/huggingface/Math-Verify

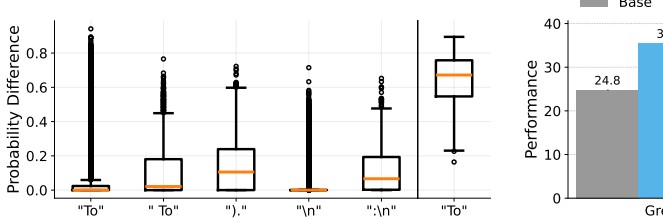

(a) Qwen2.5-Math-7B: Distribution of token-level (b) Prepending "*To*" to each prompt raises base-model probability change $\Delta P$ induced by the last-layer vec- accuracy by 10–11 points under both greedy and sam- tor over 256 DeepScaleR prompts. Five tokens with pling, capturing about 75% of the gain from the ex- the largest maxima are shown and a separate distribu- plicit last-layer vector. tion for "*To*" at the first generation position.

Figure 4: **Last-layer analysis.** Left: the last-layer vector mainly boosts the initial token "*To*". Right: prefixing that token reproduces most of the observed performance gain.

**Result.** For each layer $\ell$, we train a single steering vector $s_\ell$ while freezing all others. Figure 1 reports per-layer results for Qwen2.5-Math-7B (see Appendix A for LLaMa3.1-8B-It), compared with (i) all-layer steering, (ii) the base model with greedy decoding, and (iii) the base model sampled at $\tau = 1.0$ (the training initialization). Most layers improve over the initialization, but none matches all-layer steering; under greedy decoding, several do (Appendix B), suggesting that single-layer vectors target the right mechanisms yet cannot on their own sufficiently reduce the next-token distribution's entropy. In Qwen2.5-Math-7B, $s_{23}$ and $s_{24}$ underperform their neighboring layers; we trace the issue to vectors passing through the input layer-norm in layer 25 (Appendix C). We also find that pairing vectors improves Qwen2.5-Math-7B but offers little benefit on LLaMA3.1-8B-Instruct (Appendix D).

## 5    Steering Vector Persistence

Figure 1 shows that steering at different layers yields similar performance. We checked a hypothesis whether the steering imposed at an early layer is propagated forward and expressed by a later layer through a largely shared mechanism.

Specifically, we measured how a steering vector applied at layer $i$ persists through the network up to layer $\ell$. For each input we computed the change in hidden states after the first $\ell$ layers,

$$\Delta F_{<\ell,i}(x) = F_{<\ell}(x; s_i) - F_{<\ell}(x),$$

where $F_{<\ell}(x)$ denotes the output of the first $\ell$ layers of the transformer $F$, $s_i$ is the steering vector injected at layer $i$.

We then evaluated two quantities:

1. **Diff-Diff CosSim:** the cosine similarity between each $\Delta F_{<\ell,i}(x)$ and the mean effect $\mathbb{E}_x[\Delta F_{<\ell,i}(x)]$ over the dataset (how consistently the intervention points in the same direction).
2. **Diff-Vector CosSim:** the cosine similarity between each $\Delta F_{<\ell,i}(x)$ and the layer-$\ell$ steering vector $s_\ell$ (whether the propagated effect aligns with the layer's own steering direction).

Figure 2 summarizes the results. *Diff–Diff CosSim* shows that (a) alignment of the induced shifts gradually decays as the perturbed hidden states propagate through the network; (b) the next layer receives an almost uniform shift – cossims are always $\geq 0.8$; (c) values remain well above random (consistently $> 0.3$); and (d) the shifts on the last layers become aligned. Taken together, the shifts drift as they traverse the network but remain clustered around a common direction.

In contrast, *Diff–Vector CosSim* falls rapidly with distance from the injection layer: the propagated shifts are nearly orthogonal to each layer's own steering vector. This need not imply different behaviors – orthogonal steering directions can induce the same behavior (Jacob & Turner, 2024). Our claim is therefore mechanistic rather than behavioral: the steering implemented at different layers appears to differ in mechanism, even if the resulting behavior can coincide.

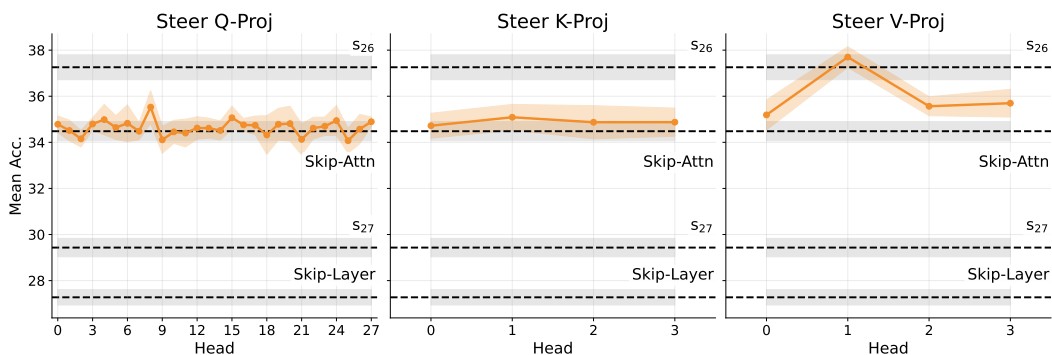

Figure 5: **Penultimate-layer steering in Qwen2.5-Math-7B.** Mean accuracy when injecting $s_{26}$ into a single projection of the final block: $Q$ (left), $K$ (center), $V$ (right). Placing $s_{26}$ only in $V_1$ closes the gap between *Skip-Attn* and $s_{26}$, indicating the effect is carried by the $V_1 \rightarrow W^O$ path and is largely independent of $Q/K$ and attention weights.

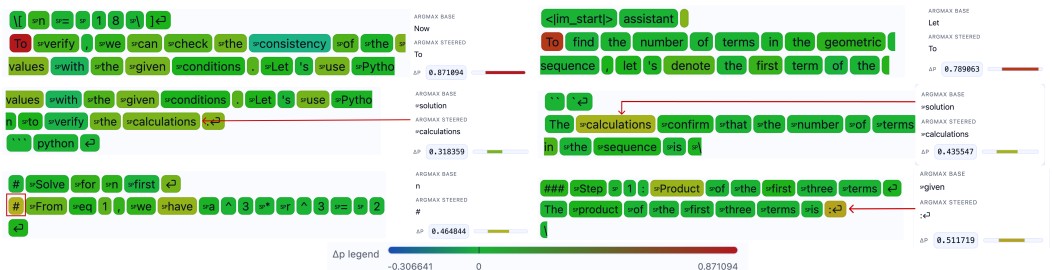

Figure 6: **Case study (Qwen2.5-Math-7B).** Token-level probability shifts ($\Delta p$) induced by *penultimate-layer* steering. Three patterns emerge: **row 1** amplify the paragraph-initial token "*To*"; **row 2** suppress "*solution*" in favor of "*calculations*"; **row 3** favor structural tokens that start Python code comments, and newlines – rather than continuing the current sentence.

Our conclusion also holds for LLAMA3.1-8B-INSTRUCT (Appendix E). The only notable difference is that *Diff–Diff CosSim* reaches the lowest point at a middle layer and then rises toward later layers, suggesting a tighter concentration around a shared steering direction in the second half of the model. See Appendix F for the raw cosine-similarity scores.

Next, note that *Diff–Diff CosSim* values jump at the final layer, indicating convergence to a more uniform effect on the unembedding. The mean shifts $\mathbb{E}[\Delta F_{<L,i}]$ produced by steering at any layer $i < L$ are highly similar to one another (Figure 3; mean pairwise cosine similarity $0.9$), so steering anywhere before the last layer yields essentially the same average bias on the unembedding. A logit-lens probe of this shared direction shows strong negative alignment with tokens in Korean, Chinese, Thai, and Arabic (cosine as low as $-0.6$; Appendix G), implying reduced probabilities for those tokens. We hypothesize that such tokens are harmful when solving English-posed math problems and are therefore down-weighted when the model is steered. In contrast, the shift from last-layer steering (equal to the steering vector itself) is clearly dissimilar to the others, suggesting a distinct mechanism that we examine next.

## 6 LAST LAYER – TOKEN SUBSTITUTION

Notice that training only the last-layer vector $s_{27}$ closes over 50% of the gap between the base model and all-layer steering, indicating a strong task signal and the reason to study it on its own. With no subsequent layers to process it, $s_{27}$ acts at unembedding without altering hidden states, effectively substituting tokens by boosting the logit of those it aligns with. We read out these preferences via a `logit-lens` projection (nostalgebraist, 2020), multiplying $s_{27}$ by the unembedding matrix (omitting the pre-unembed layer norm); the top token is *"To"* (score 42.5; cosine 0.37) (see Appendix I

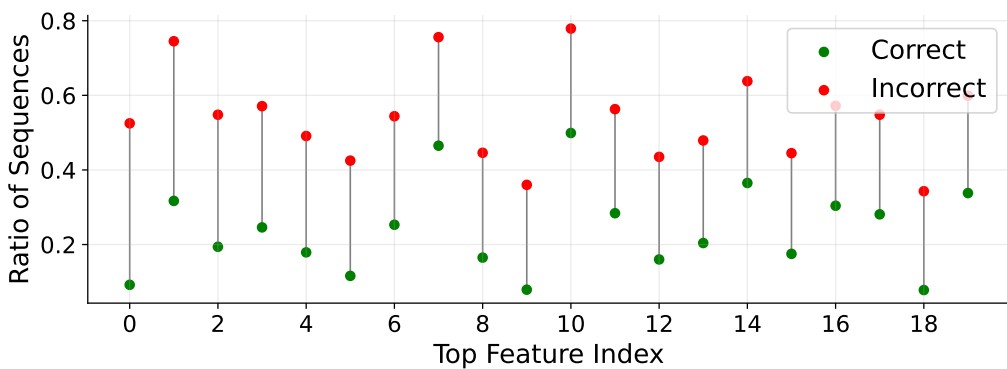

Figure 7: **Minimal CAS**. In setup $\ell = 17, \ell + k = 20$, we find features with the least values of CAS, i.e. that are the most related to incorrectness of the generation; top feature has activate in incorrect generations 5 times more frequently. Please refer to Section 8.1 for details.

for other top-10 tokens). Though the vector is added unconditionally, because softmax is nonlinear, effects vary by position, so we estimate the induced probability differences on 256 DeepScaleR prompts:

$$\Delta P_i = P(V_i \mid x_{:t}; \theta, s_{27}) - P(V_i \mid x_{:t}; \theta).$$

Grouping by token, the largest increases are for *"To"* and *" To"*, concentrated at the first generated token (Figure 4a). To test first-token steering directly, we simply append *"To"* to each prompt and evaluate the *base* model: accuracy rises by 10-11 points under both greedy and sampling – about 75% of the gain from $s_{27}$ (Figure 4b).

The last-layer vector in LLaMa3.1-8B-It has a much smaller impact (Appendix A), suggesting token substitution is less effective. Nonetheless, $s_{31}$ again concentrates its influence on the first generated token and preferentially promotes *"Step"* (Appendix J).

## 7 PENULTIMATE LAYER – CIRCUIT

Steering the penultimate layer $s_{26}$ yields a larger accuracy gain than steering the last layer, and remains tractable to analyze because the modified activations traverse only one remaining block. Here we identify which parts of that block convert the steering signal into performance.

For residual input $X$, the block computes $Y = X + \text{MHA}(\text{LN}(X))$ and $Z = Y + \text{MLP}(\text{LN}(Y))$, with heads $H_i(U) = \text{Softmax}(UW_i^Q(UW_i^K)^\top/\sqrt{d_k})\,UW_i^V$ concatenated and mixed by $W^O$, and an MLP $f(UW_1 + b_1)W_2 + b_2$. We assess the contribution of each submodule by inserting or omitting $s_{l-1}$ at specific locations and measuring mean accuracy: **Full steering** $X \leftarrow X + s_{l-1}$; **Skip-Layer** $Z \leftarrow Z + s_{l-1}$; **Skip-Attn** $Y \leftarrow Y + s_{l-1}$; **Steer-Q/K/V-Proj** for a head $i$, $(UW_i^{Q/K/V}) \mapsto (U + s_{l-1})W_i^{Q/K/V}$.

Figure 5 gives three takeaways: (i) *Skip-Layer* reduces accuracy relative to passing $s_{26}$ through the block, but the drop is small compared with using $s_{27}$; thus $s_{26}$ still helps via a direct push on the unembedding, with the remainder coming from in-block processing; (ii) *Skip-Attn* preserves over half of the $s_{26}$ gain, pointing to the MLP as the main contributor; (iii) patching any single $Q$ or $K$, or a $V_j$ with $j \neq 1$, has little effect, whereas placing $s_{26}$ only in $V_1$ closes the gap to the full $s_{26}$ result.

Viewed through the QK/OV-circuit lens (Elhage et al., 2021), token–token interaction (QK) is unchanged and the effect travels through the OV path – controlling what is written when a token is attended. Moreover, because $s_{l-1}W_1^V W_1^O$ enters the residual regardless of attention weights (Appendix K), this is equivalent to adding the projected vector just before the MLP, i.e., skipping attention. Indeed, a vector trained directly on the post-attention residual reaches $38.8 \pm 0.6$ mean accuracy, matching $s_{26}$. Overall, the penultimate vector in Qwen2.5-Math-7B acts via two routes:

Table 1: **Transferability of steering vectors across model families.** Each cell shows the mean performance change when the steering vector trained for the *Donor* model is applied to the *Recipient* model. Values are normalized so that the recipient with its own vectors equals 1.0 and the base (no vectors) equals 0.0; negative values denote degradation. "—" indicates not applicable (no Math checkpoint available).

| | | Donor | | |
| Family | Recipient | Base | Instruct | Math |
|---|---|---|---|---|
| Qwen2.5-1.5B | Base | 1.00 | 0.38 | 0.32 |
| | Instruct | 0.94 | 1.00 | 0.31 |
| | Math | 0.36 | 0.21 | 1.00 |
| Qwen2.5-7B | Base | 1.00 | 0.36 | 0.74 |
| | Instruct | 0.55 | 1.00 | -0.34 |
| | Math | 0.32 | 0.05 | 1.00 |
| LLaMA-3.1-8B | Base | 1.00 | 0.01 | — |
| | Instruct | -0.01 | 1.00 | — |

a direct effect on the unembedding and an interaction with the MLP. Appendix L contains the LLaMa3.1-8B-It study.

Figure 8 shows how adding the steering vector to the post-attention residual stream shifts token probabilities. Beyond boosting the probability of the paragraph-initial token *To*, it promotes process words – e.g., replacing "solution" with "calculations," possibly to deter premature endings. It also favors structural tokens such as Python comment markers and newlines, which often precede math blocks and may support in-code reasoning.

## 8 INTERPRETATION OF STEERING VECTORS WITH DIFFSAE

Sparse autoencoders (SAEs) decompose model activations into interpretable "features" by expressing each hidden state as a sparse weighted sum of latents from a large learned dictionary (Bricken et al., 2023). Because these latents lie in the same vector space as the model's activations, directly interpreting steering vectors with standard SAEs is infeasible (Mayne et al., 2024); we therefore focus on interpreting the *effects* of steering. Recent work has introduced methods for comparing models via their differences. In this paper, we use *DiffSAE* (Aranguri et al., 2025) – a sparse autoencoder trained to reconstruct activation differences between two models – to analyze the difference between outputs at layer $\ell + k$ of the base model and those of the model steered at layer $\ell$. Concretely, let $h_b^{(\ell+k)}, h_s^{(\ell+k)} \in \mathbb{R}^d$ be outputs of the $(\ell+k)$th layer of base and steered at layer $\ell$ models respectively. Define $d^{(\ell)} := h_s^{(\ell)} - h_b^{(\ell)}$ we train DiffSAE with reconstruction error $\mathcal{L}_{\text{rec}} = \|d - \widehat{d}\|^2 \to \min$ with auxiliary loss that forces reconstruction with features that were not activated for several batches (Gao et al., 2024). We train different DiffSAE on all setups $(\ell, \ell + k)$ for $\ell \geq 10$, i.e. model was steered at layer $\ell$, and we decompose the difference between layer $\ell + k$ outputs. See Appendix R

### 8.1 FEATURE, RELATED TO INCORRECT GENERATIONS

Our goal is to explain how steering alters the base model's behavior. To this end, we identify features that strongly correlate with producing correct solutions. We define the *correctness association score* (CAS) for feature $i$ as $\text{CAS}_i = r_i^{\text{C}} - r_i^{\text{I}}$, where $r_i^{\text{C}}$ is the fraction of correct generations in which feature $i$ activates on at least one token (among all correct generations), and $r_i^{\text{I}}$ is the analogous fraction for incorrect generations.

**Case Study: Prompt Feature** We examine the feature with the lowest CAS in the setting $\ell = 16$, $\ell + k = 20$. The top 20 features for this setup are shown in Figure 7. Surprisingly, we find features that exhibit a strong negative correlation with the correctness of the final answer.

```
system Please reason step by step , and put your final answer
within \boxed{}. <|im_end|> <|im_start|> user There are three types
of snacks for the kitten . It eats a stick of cat food every 1
day , an egg yolk every 2 days , and nutritional cream every 3
days . The kitten ate cat stick and nutritional cream on March
23 , and ate cat stick and egg yolk on March 25 . Which day in
March does the kitten eat all three types of snacks for the first
time ? <|im_end|> <|im_start|> assistant To solve this problem , we
need to find the first day in March that satisfies the condition
where the kitten eats all three types of snacks ( cat stick , egg
yolk , and nutritional cream ) on the same day . Let 's denote the
days in March as \( 1 , 2 , 3 , \ldots , 31 \). We know that :
1. The kitten eats a stick of cat food every 1 day . 2. The
```

```
system Please reason step by step , and put your final answer
within \boxed{}. <|im_end|> <|im_start|> user A travel agency has
120 rooms , with a daily rental rate of 50 yuan per room , and
they are fully booked every day . After renovation , the agency
plans to increase the rental rate . Market research indicates that
for every 5 yuan increase in the daily rental rate per room ,
the number of rooms rented out each day will decrease by 6 .
Without considering other factors , to what amount should the
agency increase the daily rental rate per room to maximize the
total daily rental income ? And how much more will this be
compared to the total daily rental income before the renovation ?
<|im_end|> <|im_start|> assistant To solve this problem , we need
to find the optimal rental rate that maximizes the total daily
```

```
system Please reason step by step , and put your final answer
within \boxed{}. <|im_end|> <|im_start|> user Given the function
$f(x)=\ln x+\frac{1}{2}x^2-ax$, where $a \in
\mathbb{R}$, it has two extreme points at $x=x_1$ and
$x=x_2$, with $x_1 < x_2$. (I) When $a=3$, find the
extreme values of the function $f(x)$. (II) If
$x_2\ge s_1$ ant ex_1$ ($e$ is the base of the natural
logarithm ), find the maximum value of $f(x_2)-f(x_1)$.
<|im_end|> <|im_start|> assistant cost Assistant : Let 's break
down the problem into two parts and solve it step by step using
Python and SymPy . ### Part (I): When $a=3$, find the extreme
values of the function $f(x)$. 1. ** Find the derivative of
$f(x)$: ** \[ f'(x)=\frac{d}{dx}\left(\ln x+
```

Figure 8: Top-1 feature by absolute frequency diff. We suspect that this feature indicates challenging task for the model, and model might know if it could solve it before generation. See Subsection 8.1 for more details.

We focus on the feature with the largest frequency difference: it activates on approximately 60% of incorrect answers but only on about 10% of correct answers. Examples of these activations are presented in Figure 8. The feature tends to fire on the task description, which may suggest that the model has an early indication of whether it can solve the task before generating an answer. Additional features and their descriptions are provided in Appendix Q.

## 9 TRANSFER OF STEERING VECTORS ACROSS MODELS

We test whether steering vectors learned for one model can improve another model from the same model family. We consider three groups of checkpoints models: (i) {Qwen2.5-7B, Qwen2.5-7B-Instruct, Qwen2.5-Math-7B}, (ii) {Qwen2.5-1.5B, Qwen2.5-1.5B-Instruct, Qwen2.5-Math-1.5B}, and (iii) {LLaMA-3.1-8B, LLaMA-3.1-8B-Instruct} – where models within a group share hidden size and depth. For each ordered pair within a group, we swap the donor model's steering vectors into the recipient and report the *relative* gain: scores are normalized by the gap between the base model and the same model equipped with its own vectors. Raw (unnormalized) scores are provided in Appendix M.

Table 1 summarizes the results. Transfers between Qwen2.5-7B-Instruct and Qwen2.5-Math-7B are weak and sometimes harmful, suggesting their fine-tuning objectives induce incompatible directions. In contrast, all other exchanges yield positive gains, with the base Qwen2.5 checkpoints consistently serving as the strongest donors for both 7B and 1.5B recipients. In contrast, the models from LLaMa3.1-8B family are unaffected by the transferred steering vectors showing no change in performance. We attribute this result to the fact that the two models use different chat templates (Appendix O) and steering vectors trained on one template do not have an effect on another. It is surprising, however, that such transfer does not deteriorate the performance of the recipient model, which deserves further study.

Overall, this experiments provides mixed results. indicate that the latent directions associated with strong reasoning ability are largely preserved after fine-tuning. Consequently, reusing steering vectors trained on a base model can be a simple, low-cost way to lift performance on related checkpoints and domains.

## 10 ADAPTIVE STEERING

We next ask how allowing token-conditional magnitude affects steering performance, and which tokens are amplified or suppressed.

We implement adaptive-magnitude steering with a rank-1 LoRA on the MLP output matrix of each layer. With LoRA, the MLP output becomes $W^{n,m}x^m + B^{n,1}A^{1,m}x^m$, where we treat $B$ as the steering direction and $Ax$ as its token-conditioned magnitude, computed from the hidden state $x$.

We train this rank-1 LoRA separately at each layer of Qwen2.5-Math-7B and LLaMA3.1-8B-Instruct, following the setup in Section 4. As shown in Figure 9, adaptive steering consistently outperforms constant-magnitude steering and matches full steering on the first 20 layers.

To explain the activation magnitudes, we examine activation patterns and run an automatic-interpretability style pipeline (Bills et al., 2023). For each DeepScaleR prompt and each layer-$i$ LoRA, we generate a completion and record token-wise magnitudes across 100 examples (see Figure 25).

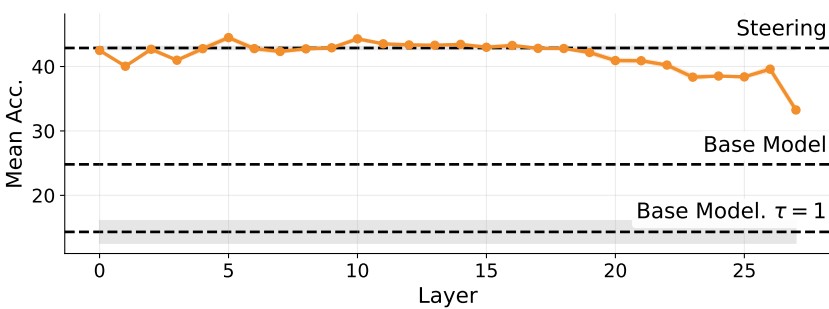

Figure 9: **Single-layer adaptive steering.** Mean accuracy on six benchmarks for Qwen2.5-Math-7B when training a single steering vector with token-dependent magnitude at layer $\ell$ with all other layers frozen. Adaptive steering always surpasses the constant steering a matches the performance of full-steering on the 20 first layers.

We then pass either one labeled example or a batch of them to the model and ask it to describe what drives the activation. Examples are given in Appendix S. Two main trends emerge:

**(i) Layerwise shift.** Early-layer LoRAs fire on solution steps and formatting that set the answer's structure – more syntactic and structural cues. Later layers shift toward identifying reasoning steps and the high-level semantic structure of the answer, focusing less on surface form and more on the underlying reasoning. This aligns with the transformer's progression from local aggregation to higher-level representations.

**(ii) Positional dependence.** Activations depend strongly on where a token appears: instruction, reasoning, or final answer. For example, in Figure 26, the layer-15 LoRA is negative on instruction and answer tokens (e.g., "proceed," "following," "therefore"), but positive on tokens such as "when" and "which" within the reasoning span.

## 11 CONCLUSION

We presented a mechanistic interpretation of steering vectors trained for mathematical reasoning. These vectors (i) suppress specific token groups, (ii) substitute useful opening tokens, (iii) achieve strong effects without relying on attention, acting largely through the MLP; and (iv) are both composable and transferable.

Though most our findings are consistent across the two models, several effects are clear on Qwen but markedly weaker on LLaMA, suggesting model-specific mechanisms. Systematic comparisons across architectures may help explain observed performance and behavioural differences. A promising direction is to pin down the precise mechanism of mid-layer steering vectors.

Overall, steering vectors provide a compact and informative probe of reasoning-trained models, offering concrete insight into the changes induced by such training.

## ETHICS STATEMENT

Interpretability research can be dual-use, but our study remains within the safety bounds of the underlying models and aims to advance responsible AI through improved model understanding. We analyze Qwen and LLaMA models using the DeepScaler dataset, which excludes known harmful content. We are transparent about limitations and report no conflicts of interest.

## REPRODUCIBILITY STATEMENT

We aim for strong reproducibility. Our experiments use publicly available models (Qwen and LLaMA families), the DeepScaleR training dataset, and standard math benchmarks (AIME24/25, AMC23, MATH500, OlympiadBench, and MinervaMath). Section 4 and the appendices provide detailed

descriptions of the SAE training procedure, hyperparameter settings and raw evaluation numbers. We will open-source the full implementation – including training code and analysis scripts – to facilitate replication.

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

## A    SINGLE-LAYER LLAMA3.1-8B-IT

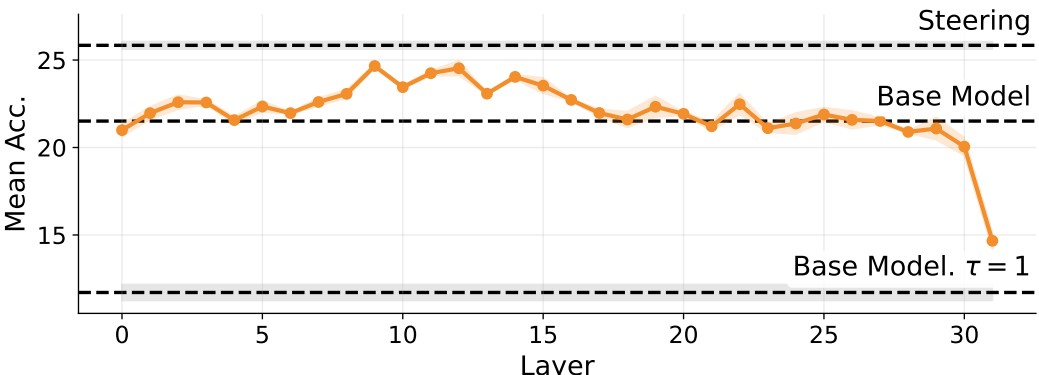

Figure 10: **Single-layer steering.** Mean accuracy on six benchmarks for Qwen2.5-Math-7B when training a single vector $s_\ell$ at layer $\ell$ with all other layers frozen. Vectors from layers $8 - 15$ yield the largest gains but never match all-layer steering, indicating the improvement is distributed across layers.

The results for LLaMa3.1-8B-It in Figure 10 mirror those for Qwen2.5-Math-7B in Section 4: mid-layer vectors perform best yet none reaches all-layer steering. Differently from Qwen2.5-Math-7B, the final-layer vector yields only marginal gains.

## B    PER-LAYER STEERING WITH GREEDY DECODING

We evaluated the same single-layer steering vectors from Section 4 with temperature $\tau = 0$ and found that many match the performance of full-steering in this setup, see Figure 11. That shows that single-layer steering vectors modify the right mechanisms but lack the capacity to lower the entropy enough.

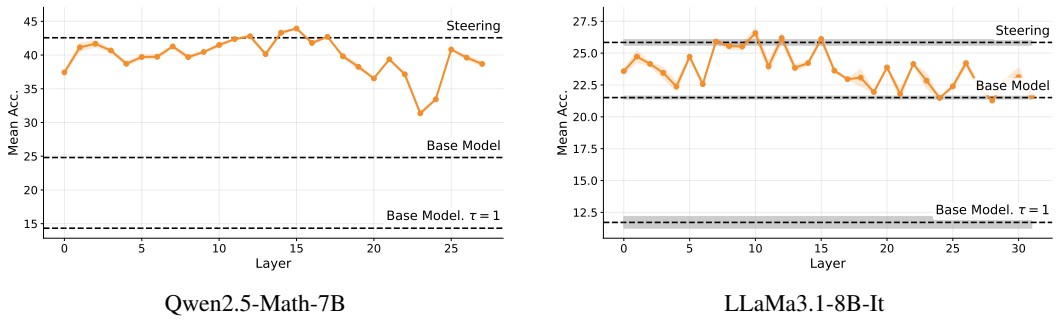

Qwen2.5-Math-7B                      LLaMa3.1-8B-It

Figure 11: **Single-Layer Steering with** $\tau = 0$.

We re-evaluate the single-layer steering vectors from Section 4 with greedy decoding ($\tau = 0$). As shown in Figure 11, many of these single-layer vectors match the performance of full steering, which we did not observe when sampling with temperature $\tau = 1$. This suggests they target the correct mechanisms but lack the capacity to sufficiently reduce generation entropy.

## C   INEFFECTIVE LAYERS IN QWEN2.5-MATH-7B

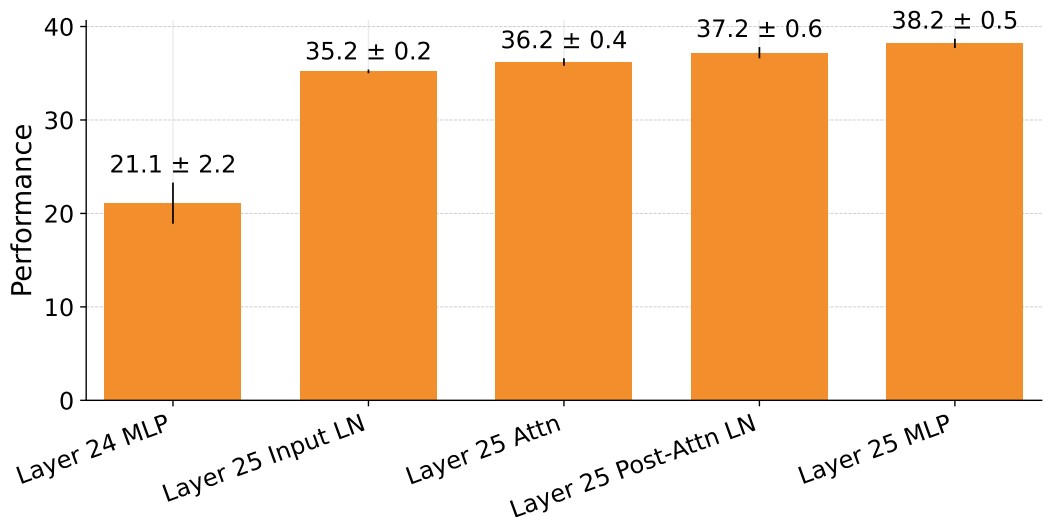

Figure 12: **Component-output steering.**

As noted in Section 4, single-layer steering on layers 23 and 24 underperforms their neighbors. To pinpoint where this loss arises, we trained vectors inserted immediately after each subcomponent between the layer-24 MLP and the layer-25 MLP. Figure 12 shows that placing $s_{24}$ *after* the input LayerNorm of layer 25 closes the gap with $s_{25}$. Thus the input LayerNorm is the problematic step – passing through it limits the effect of the steering vector.

## D   STEERING VECTORS ARE COMPOSABLE

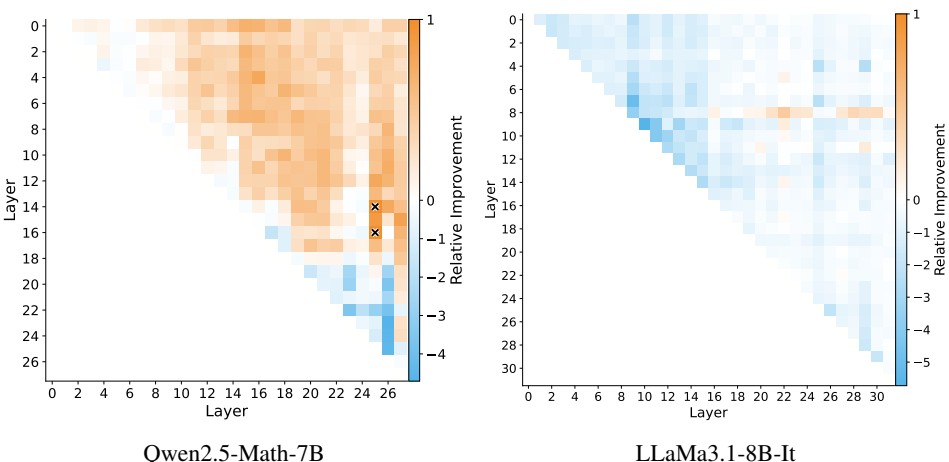

Figure 13: **Composable steering.** The normalized gain in mean accuracy when pairing vectors $s_i$ and $s_j$ with $i < j$. Crosses mark pairs reaching $\geq 99\%$ of all-layer. Qwen2.5-Math-7B often benefits, with two near–all-layer pairs; LLaMa3.1-8B-It is mostly neutral or interfering.

We test whether depth-specific vectors combine without conflict by evaluating all pairs $(s_i, s_j)$ with $i < j$. Figure 13 reports the normalized gain

$$\text{norm}(s_i, s_j) = \frac{\text{Acc}(s_i, s_j) - \max\{\text{Acc}(s_i),\ \text{Acc}(s_j)\}}{\text{Acc}(\mathbb{S}) - \max\{\text{Acc}(s_i),\ \text{Acc}(s_j)\}},$$

which compares the pair to the better single vector: $0$ means no improvement, $1$ matches the all-layer score $\mathbb{S}$, and $< 0$ indicates interference. Exact accuracies are in Appendix N.

Adjacent pairs (near the diagonal) often interfere, while wider gaps add constructively. Notably, $s_{25}$ paired with $s_{16}$ or $s_{14}$ nearly matches the all-layer $42.9\%$, though each alone plateaus around $40\%$. The composition in LLaMa3.1-8B-It is weaker: many pairs are neutral or harmful. The modest gains concentrate when late layers are paired with mid-depth layer $8$, but none reach the all-layer result.

# E    STEERING VECTOR PERSISTENCE. LLaMa3.1-8B-It

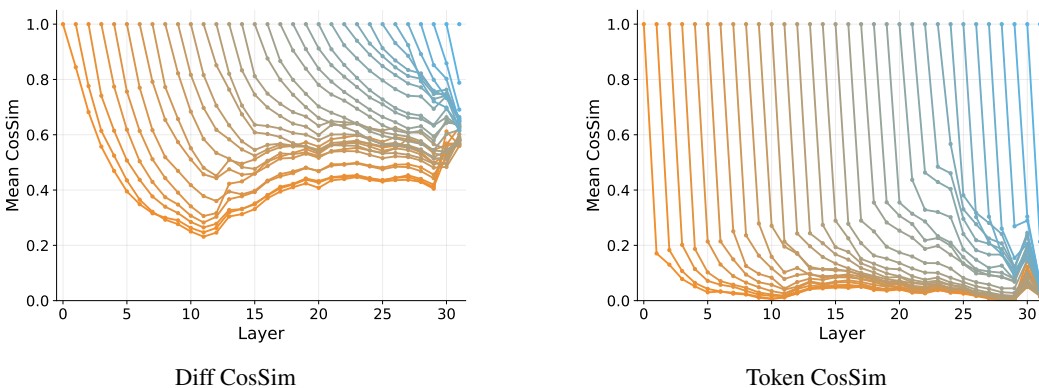

Diff CosSim                                    Token CosSim

Figure 14: **Steering Vector Persistence – LLaMa3.1-8B-It.** For each steering vector injection layer $k$ (one colored curve per $k$; warm = early layers, cool = later layers) we plot, as a function of target layer $l$ (x-axis), the mean cosine similarity of the per-token change in hidden state $\Delta F_{<l}(x)$. *Left*: similarity between each $\Delta F_{<l}(x)$ and the dataset mean $\mathbb{E}[\Delta F_{<l}(x)]$. *Right*: similarity between $\Delta F_{<l}(x)$ and the layer-$l$ steering vector $s_l$.

# F   STEERING VECTOR PERSISTENCE. RAW NUMBERS

Table 2: **Qwen2.5-Math-7B.** Raw scores for the plots in Figure 2.

### Diff-Diff CosSim

| Layer | 0 | 1 | 2 | 3 | 4 | 5 | 6 | 7 | 8 | 9 | 10 | 11 | 12 | 13 | 14 | 15 | 16 | 17 | 18 | 19 | 20 | 21 | 22 | 23 | 24 | 25 | 26 | 27 |
|---|---|---|---|---|---|---|---|---|---|---|---|---|---|---|---|---|---|---|---|---|---|---|---|---|---|---|---|---|
| 0 | 1.00 | 0.96 | 0.91 | 0.84 | 0.80 | 0.75 | 0.67 | 0.59 | 0.54 | 0.54 | 0.52 | 0.49 | 0.47 | 0.45 | 0.43 | 0.42 | 0.40 | 0.37 | 0.37 | 0.37 | 0.35 | 0.34 | 0.34 | 0.33 | 0.33 | 0.31 | 0.37 | 0.57 |
| 1 | — | 1.00 | 0.95 | 0.85 | 0.80 | 0.74 | 0.65 | 0.58 | 0.51 | 0.51 | 0.49 | 0.45 | 0.43 | 0.42 | 0.39 | 0.38 | 0.36 | 0.34 | 0.33 | 0.33 | 0.31 | 0.32 | 0.32 | 0.31 | 0.31 | 0.29 | 0.29 | 0.41 |
| 2 | — | — | 1.00 | 0.96 | 0.91 | 0.86 | 0.78 | 0.70 | 0.64 | 0.64 | 0.61 | 0.57 | 0.54 | 0.52 | 0.50 | 0.48 | 0.45 | 0.42 | 0.42 | 0.41 | 0.39 | 0.37 | 0.37 | 0.37 | 0.36 | 0.34 | 0.38 | 0.55 |
| 3 | — | — | — | 1.00 | 0.93 | 0.87 | 0.78 | 0.69 | 0.64 | 0.63 | 0.61 | 0.57 | 0.55 | 0.53 | 0.51 | 0.49 | 0.47 | 0.44 | 0.43 | 0.41 | 0.39 | 0.37 | 0.35 | 0.34 | 0.32 | 0.30 | 0.32 | 0.57 |
| 4 | — | — | — | — | 1.00 | 0.92 | 0.82 | 0.73 | 0.67 | 0.67 | 0.64 | 0.60 | 0.58 | 0.56 | 0.53 | 0.51 | 0.49 | 0.46 | 0.45 | 0.44 | 0.41 | 0.39 | 0.37 | 0.36 | 0.34 | 0.32 | 0.34 | 0.61 |
| 5 | — | — | — | — | — | 1.00 | 0.88 | 0.77 | 0.71 | 0.69 | 0.66 | 0.62 | 0.59 | 0.56 | 0.53 | 0.51 | 0.49 | 0.46 | 0.44 | 0.43 | 0.40 | 0.38 | 0.37 | 0.36 | 0.35 | 0.32 | 0.36 | 0.62 |
| 6 | — | — | — | — | — | — | 1.00 | 0.82 | 0.72 | 0.70 | 0.65 | 0.61 | 0.58 | 0.55 | 0.52 | 0.50 | 0.47 | 0.44 | 0.44 | 0.42 | 0.39 | 0.38 | 0.36 | 0.35 | 0.33 | 0.32 | 0.33 | 0.59 |
| 7 | — | — | — | — | — | — | — | 1.00 | 0.85 | 0.81 | 0.74 | 0.67 | 0.63 | 0.60 | 0.56 | 0.54 | 0.51 | 0.47 | 0.46 | 0.45 | 0.41 | 0.39 | 0.38 | 0.37 | 0.35 | 0.33 | 0.36 | 0.64 |
| 8 | — | — | — | — | — | — | — | — | 1.00 | 0.87 | 0.77 | 0.69 | 0.64 | 0.60 | 0.56 | 0.54 | 0.51 | 0.47 | 0.46 | 0.45 | 0.41 | 0.40 | 0.38 | 0.37 | 0.36 | 0.33 | 0.36 | 0.64 |
| 9 | — | — | — | — | — | — | — | — | — | 1.00 | 0.85 | 0.75 | 0.70 | 0.64 | 0.59 | 0.55 | 0.52 | 0.48 | 0.47 | 0.45 | 0.42 | 0.41 | 0.39 | 0.39 | 0.37 | 0.35 | 0.37 | 0.68 |
| 10 | — | — | — | — | — | — | — | — | — | — | 1.00 | 0.85 | 0.76 | 0.69 | 0.64 | 0.60 | 0.57 | 0.52 | 0.51 | 0.49 | 0.46 | 0.45 | 0.43 | 0.42 | 0.39 | 0.37 | 0.40 | 0.70 |
| 11 | — | — | — | — | — | — | — | — | — | — | — | 1.00 | 0.84 | 0.75 | 0.67 | 0.63 | 0.58 | 0.53 | 0.51 | 0.51 | 0.48 | 0.47 | 0.45 | 0.43 | 0.41 | 0.39 | 0.40 | 0.68 |
| 12 | — | — | — | — | — | — | — | — | — | — | — | — | 1.00 | 0.85 | 0.76 | 0.71 | 0.66 | 0.58 | 0.56 | 0.55 | 0.52 | 0.50 | 0.48 | 0.48 | 0.45 | 0.44 | 0.44 | 0.72 |
| 13 | — | — | — | — | — | — | — | — | — | — | — | — | — | 1.00 | 0.86 | 0.78 | 0.72 | 0.63 | 0.60 | 0.60 | 0.58 | 0.56 | 0.54 | 0.52 | 0.49 | 0.47 | 0.48 | 0.73 |
| 14 | — | — | — | — | — | — | — | — | — | — | — | — | — | — | 1.00 | 0.86 | 0.77 | 0.66 | 0.63 | 0.63 | 0.60 | 0.59 | 0.56 | 0.55 | 0.51 | 0.50 | 0.50 | 0.71 |
| 15 | — | — | — | — | — | — | — | — | — | — | — | — | — | — | — | 1.00 | 0.85 | 0.74 | 0.68 | 0.67 | 0.64 | 0.62 | 0.59 | 0.59 | 0.55 | 0.55 | 0.56 | 0.75 |
| 16 | — | — | — | — | — | — | — | — | — | — | — | — | — | — | — | — | 1.00 | 0.80 | 0.73 | 0.72 | 0.69 | 0.68 | 0.64 | 0.65 | 0.62 | 0.59 | 0.58 | 0.70 |
| 17 | — | — | — | — | — | — | — | — | — | — | — | — | — | — | — | — | — | 1.00 | 0.81 | 0.73 | 0.67 | 0.64 | 0.60 | 0.62 | 0.58 | 0.57 | 0.55 | 0.57 |
| 18 | — | — | — | — | — | — | — | — | — | — | — | — | — | — | — | — | — | — | 1.00 | 0.87 | 0.79 | 0.74 | 0.68 | 0.66 | 0.62 | 0.63 | 0.67 | 0.71 |
| 19 | — | — | — | — | — | — | — | — | — | — | — | — | — | — | — | — | — | — | — | 1.00 | 0.88 | 0.81 | 0.74 | 0.70 | 0.66 | 0.67 | 0.71 | 0.75 |
| 20 | — | — | — | — | — | — | — | — | — | — | — | — | — | — | — | — | — | — | — | — | 1.00 | 0.90 | 0.83 | 0.78 | 0.72 | 0.72 | 0.72 | 0.76 |
| 21 | — | — | — | — | — | — | — | — | — | — | — | — | — | — | — | — | — | — | — | — | — | 1.00 | 0.90 | 0.85 | 0.79 | 0.75 | 0.72 | 0.81 |
| 22 | — | — | — | — | — | — | — | — | — | — | — | — | — | — | — | — | — | — | — | — | — | — | 1.00 | 0.93 | 0.86 | 0.81 | 0.72 | 0.66 |
| 23 | — | — | — | — | — | — | — | — | — | — | — | — | — | — | — | — | — | — | — | — | — | — | — | 1.00 | 0.90 | 0.81 | 0.71 | 0.55 |
| 24 | — | — | — | — | — | — | — | — | — | — | — | — | — | — | — | — | — | — | — | — | — | — | — | — | 1.00 | 0.83 | 0.74 | 0.49 |
| 25 | — | — | — | — | — | — | — | — | — | — | — | — | — | — | — | — | — | — | — | — | — | — | — | — | — | 1.00 | 0.90 | 0.92 |
| 26 | — | — | — | — | — | — | — | — | — | — | — | — | — | — | — | — | — | — | — | — | — | — | — | — | — | — | 1.00 | 0.92 |
| 27 | — | — | — | — | — | — | — | — | — | — | — | — | — | — | — | — | — | — | — | — | — | — | — | — | — | — | — | 1.00 |

### Diff-Vector CosSim

| Layer | 0 | 1 | 2 | 3 | 4 | 5 | 6 | 7 | 8 | 9 | 10 | 11 | 12 | 13 | 14 | 15 | 16 | 17 | 18 | 19 | 20 | 21 | 22 | 23 | 24 | 25 | 26 | 27 |
|---|---|---|---|---|---|---|---|---|---|---|---|---|---|---|---|---|---|---|---|---|---|---|---|---|---|---|---|---|
| 0 | 1.00 | 0.31 | 0.33 | 0.29 | 0.24 | 0.23 | 0.18 | 0.14 | 0.12 | 0.11 | 0.11 | 0.10 | 0.10 | 0.09 | 0.08 | 0.07 | 0.07 | 0.05 | 0.04 | 0.04 | 0.03 | 0.03 | 0.04 | 0.02 | 0.02 | 0.02 | 0.04 | 0.12 |
| 1 | — | 1.00 | 0.33 | 0.30 | 0.17 | 0.16 | 0.12 | 0.09 | 0.09 | 0.08 | 0.08 | 0.08 | 0.07 | 0.07 | 0.06 | 0.06 | 0.06 | 0.04 | 0.03 | 0.03 | 0.02 | 0.03 | 0.04 | 0.03 | 0.03 | 0.02 | 0.03 | 0.08 |
| 2 | — | — | 1.00 | 0.55 | 0.47 | 0.38 | 0.30 | 0.23 | 0.19 | 0.18 | 0.18 | 0.15 | 0.14 | 0.13 | 0.11 | 0.10 | 0.09 | 0.07 | 0.07 | 0.05 | 0.04 | 0.05 | 0.04 | 0.05 | 0.03 | 0.02 | 0.03 | 0.11 |
| 3 | — | — | — | 1.00 | 0.56 | 0.44 | 0.32 | 0.26 | 0.23 | 0.22 | 0.21 | 0.18 | 0.17 | 0.16 | 0.14 | 0.13 | 0.11 | 0.08 | 0.08 | 0.07 | 0.05 | 0.05 | 0.06 | 0.05 | 0.05 | 0.03 | 0.05 | 0.12 |
| 4 | — | — | — | — | 1.00 | 0.51 | 0.36 | 0.29 | 0.26 | 0.24 | 0.23 | 0.19 | 0.18 | 0.16 | 0.14 | 0.13 | 0.12 | 0.09 | 0.08 | 0.07 | 0.06 | 0.06 | 0.06 | 0.05 | 0.05 | 0.03 | 0.05 | 0.13 |
| 5 | — | — | — | — | — | 1.00 | 0.43 | 0.32 | 0.26 | 0.25 | 0.23 | 0.20 | 0.17 | 0.16 | 0.13 | 0.13 | 0.12 | 0.08 | 0.08 | 0.06 | 0.05 | 0.05 | 0.06 | 0.05 | 0.05 | 0.03 | 0.05 | 0.13 |
| 6 | — | — | — | — | — | — | 1.00 | 0.43 | 0.32 | 0.28 | 0.25 | 0.22 | 0.19 | 0.18 | 0.14 | 0.13 | 0.12 | 0.09 | 0.08 | 0.07 | 0.05 | 0.05 | 0.06 | 0.05 | 0.04 | 0.03 | 0.04 | 0.12 |
| 7 | — | — | — | — | — | — | — | 1.00 | 0.48 | 0.40 | 0.34 | 0.28 | 0.24 | 0.21 | 0.17 | 0.16 | 0.14 | 0.10 | 0.09 | 0.08 | 0.06 | 0.06 | 0.07 | 0.06 | 0.05 | 0.03 | 0.05 | 0.14 |
| 8 | — | — | — | — | — | — | — | — | 1.00 | 0.47 | 0.37 | 0.30 | 0.25 | 0.22 | 0.18 | 0.16 | 0.14 | 0.10 | 0.09 | 0.08 | 0.05 | 0.06 | 0.06 | 0.06 | 0.05 | 0.03 | 0.05 | 0.14 |
| 9 | — | — | — | — | — | — | — | — | — | 1.00 | 0.47 | 0.36 | 0.29 | 0.26 | 0.20 | 0.18 | 0.16 | 0.11 | 0.10 | 0.08 | 0.07 | 0.06 | 0.07 | 0.06 | 0.05 | 0.04 | 0.05 | 0.14 |
| 10 | — | — | — | — | — | — | — | — | — | — | 1.00 | 0.51 | 0.38 | 0.32 | 0.24 | 0.21 | 0.18 | 0.13 | 0.12 | 0.09 | 0.07 | 0.07 | 0.07 | 0.07 | 0.07 | 0.04 | 0.06 | 0.15 |
| 11 | — | — | — | — | — | — | — | — | — | — | — | 1.00 | 0.49 | 0.39 | 0.29 | 0.26 | 0.21 | 0.13 | 0.12 | 0.10 | 0.07 | 0.07 | 0.08 | 0.08 | 0.07 | 0.04 | 0.07 | 0.14 |
| 12 | — | — | — | — | — | — | — | — | — | — | — | — | 1.00 | 0.56 | 0.40 | 0.31 | 0.24 | 0.16 | 0.13 | 0.10 | 0.07 | 0.07 | 0.08 | 0.08 | 0.07 | 0.04 | 0.07 | 0.15 |
| 13 | — | — | — | — | — | — | — | — | — | — | — | — | — | 1.00 | 0.55 | 0.41 | 0.30 | 0.20 | 0.16 | 0.12 | 0.09 | 0.09 | 0.08 | 0.09 | 0.08 | 0.05 | 0.08 | 0.15 |
| 14 | — | — | — | — | — | — | — | — | — | — | — | — | — | — | 1.00 | 0.53 | 0.37 | 0.23 | 0.18 | 0.13 | 0.11 | 0.10 | 0.09 | 0.09 | 0.09 | 0.05 | 0.09 | 0.15 |
| 15 | — | — | — | — | — | — | — | — | — | — | — | — | — | — | — | 1.00 | 0.55 | 0.35 | 0.27 | 0.19 | 0.15 | 0.12 | 0.12 | 0.13 | 0.12 | 0.06 | 0.12 | 0.16 |
| 16 | — | — | — | — | — | — | — | — | — | — | — | — | — | — | — | — | 1.00 | 0.44 | 0.29 | 0.19 | 0.13 | 0.11 | 0.10 | 0.11 | 0.10 | 0.05 | 0.10 | 0.15 |
| 17 | — | — | — | — | — | — | — | — | — | — | — | — | — | — | — | — | — | 1.00 | 0.33 | 0.18 | 0.13 | 0.11 | 0.09 | 0.10 | 0.10 | 0.04 | 0.08 | 0.11 |
| 18 | — | — | — | — | — | — | — | — | — | — | — | — | — | — | — | — | — | — | 1.00 | 0.36 | 0.28 | 0.20 | 0.19 | 0.21 | 0.19 | 0.10 | 0.20 | 0.13 |
| 19 | — | — | — | — | — | — | — | — | — | — | — | — | — | — | — | — | — | — | — | 1.00 | 0.42 | 0.27 | 0.24 | 0.26 | 0.23 | 0.13 | 0.24 | 0.14 |
| 20 | — | — | — | — | — | — | — | — | — | — | — | — | — | — | — | — | — | — | — | — | 1.00 | 0.41 | 0.36 | 0.35 | 0.30 | 0.18 | 0.26 | 0.15 |
| 21 | — | — | — | — | — | — | — | — | — | — | — | — | — | — | — | — | — | — | — | — | — | 1.00 | 0.38 | 0.33 | 0.28 | 0.24 | 0.29 | 0.17 |
| 22 | — | — | — | — | — | — | — | — | — | — | — | — | — | — | — | — | — | — | — | — | — | — | 1.00 | 0.48 | 0.37 | 0.32 | 0.22 | 0.15 |
| 23 | — | — | — | — | — | — | — | — | — | — | — | — | — | — | — | — | — | — | — | — | — | — | — | 1.00 | 0.68 | 0.17 | 0.22 | 0.10 |
| 24 | — | — | — | — | — | — | — | — | — | — | — | — | — | — | — | — | — | — | — | — | — | — | — | — | 1.00 | 0.17 | 0.22 | 0.07 |
| 25 | — | — | — | — | — | — | — | — | — | — | — | — | — | — | — | — | — | — | — | — | — | — | — | — | — | 1.00 | 0.48 | 0.23 |
| 26 | — | — | — | — | — | — | — | — | — | — | — | — | — | — | — | — | — | — | — | — | — | — | — | — | — | — | 1.00 | 0.24 |
| 27 | — | — | — | — | — | — | — | — | — | — | — | — | — | — | — | — | — | — | — | — | — | — | — | — | — | — | — | 1.00 |

Table 3: **LLaMa3.1-8B-It.** Raw scores for the plots in Figure 14.

Diff-Diff CosSim

| Layer | 0 | 1 | 2 | 3 | 4 | 5 | 6 | 7 | 8 | 9 | 10 | 11 | 12 | 13 | 14 | 15 | 16 | 17 | 18 | 19 | 20 | 21 | 22 | 23 | 24 | 25 | 26 | 27 | 28 | 29 | 30 | 31 |
|---|---|---|---|---|---|---|---|---|---|---|---|---|---|---|---|---|---|---|---|---|---|---|---|---|---|---|---|---|---|---|---|---|
| 0 | 1.00 | 0.84 | 0.68 | 0.55 | 0.46 | 0.38 | 0.34 | 0.31 | 0.30 | 0.29 | 0.26 | 0.24 | 0.27 | 0.34 | 0.35 | 0.37 | 0.40 | 0.42 | 0.43 | 0.46 | 0.44 | 0.46 | 0.46 | 0.45 | 0.45 | 0.46 | 0.46 | 0.44 | 0.42 | 0.53 | 0.57 | |
| 1 | — | 1.00 | 0.77 | 0.61 | 0.51 | 0.42 | 0.36 | 0.32 | 0.29 | 0.28 | 0.25 | 0.24 | 0.25 | 0.33 | 0.34 | 0.36 | 0.40 | 0.43 | 0.45 | 0.47 | 0.45 | 0.48 | 0.48 | 0.49 | 0.48 | 0.47 | 0.48 | 0.48 | 0.47 | 0.44 | 0.62 | 0.57 |
| 2 | — | — | 1.00 | 0.74 | 0.61 | 0.50 | 0.44 | 0.38 | 0.34 | 0.32 | 0.29 | 0.27 | 0.28 | 0.33 | 0.33 | 0.36 | 0.38 | 0.41 | 0.44 | 0.43 | 0.45 | 0.45 | 0.45 | 0.44 | 0.44 | 0.44 | 0.46 | 0.45 | 0.43 | 0.58 | 0.56 | |
| 3 | — | — | — | 1.00 | 0.78 | 0.62 | 0.51 | 0.44 | 0.38 | 0.35 | 0.31 | 0.29 | 0.31 | 0.37 | 0.39 | 0.40 | 0.45 | 0.46 | 0.47 | 0.49 | 0.48 | 0.50 | 0.50 | 0.51 | 0.50 | 0.49 | 0.50 | 0.50 | 0.50 | 0.47 | 0.54 | 0.58 |
| 4 | — | — | — | — | 1.00 | 0.75 | 0.61 | 0.50 | 0.44 | 0.38 | 0.33 | 0.30 | 0.31 | 0.43 | 0.44 | 0.47 | 0.51 | 0.52 | 0.53 | 0.55 | 0.53 | 0.56 | 0.56 | 0.58 | 0.56 | 0.55 | 0.56 | 0.55 | 0.54 | 0.50 | 0.53 | 0.53 |
| 5 | — | — | — | — | — | 1.00 | 0.77 | 0.62 | 0.52 | 0.47 | 0.42 | 0.38 | 0.36 | 0.39 | 0.37 | 0.38 | 0.42 | 0.44 | 0.45 | 0.46 | 0.45 | 0.47 | 0.47 | 0.48 | 0.47 | 0.46 | 0.47 | 0.47 | 0.46 | 0.43 | 0.55 | 0.57 |
| 6 | — | — | — | — | — | — | 1.00 | 0.80 | 0.66 | 0.58 | 0.52 | 0.47 | 0.46 | 0.47 | 0.45 | 0.46 | 0.48 | 0.51 | 0.51 | 0.52 | 0.51 | 0.53 | 0.54 | 0.53 | 0.54 | 0.59 | 0.57 | 0.55 | 0.58 | 0.64 | | |
| 7 | — | — | — | — | — | — | — | 1.00 | 0.77 | 0.66 | 0.55 | 0.48 | 0.45 | 0.51 | 0.50 | 0.52 | 0.54 | 0.56 | 0.57 | 0.58 | 0.56 | 0.59 | 0.59 | 0.60 | 0.59 | 0.57 | 0.60 | 0.59 | 0.58 | 0.54 | 0.53 | 0.56 |
| 8 | — | — | — | — | — | — | — | — | 1.00 | 0.82 | 0.68 | 0.58 | 0.53 | 0.51 | 0.48 | 0.47 | 0.51 | 0.53 | 0.53 | 0.54 | 0.52 | 0.54 | 0.52 | 0.52 | 0.51 | 0.48 | 0.48 | 0.55 | | | | |
| 9 | — | — | — | — | — | — | — | — | — | 1.00 | 0.82 | 0.70 | 0.62 | 0.58 | 0.55 | 0.53 | 0.55 | 0.57 | 0.56 | 0.56 | 0.55 | 0.57 | 0.57 | 0.58 | 0.56 | 0.55 | 0.55 | 0.55 | 0.53 | 0.50 | 0.50 | 0.59 |
| 10 | — | — | — | — | — | — | — | — | — | — | 1.00 | 0.82 | 0.71 | 0.64 | 0.59 | 0.56 | 0.58 | 0.59 | 0.58 | 0.58 | 0.56 | 0.59 | 0.59 | 0.60 | 0.58 | 0.59 | 0.58 | 0.57 | 0.54 | 0.54 | 0.62 | |
| 11 | — | — | — | — | — | — | — | — | — | — | — | 1.00 | 0.85 | 0.76 | 0.67 | 0.60 | 0.60 | 0.59 | 0.57 | 0.55 | 0.53 | 0.56 | 0.56 | 0.56 | 0.54 | 0.53 | 0.53 | 0.52 | 0.51 | 0.49 | 0.50 | 0.59 |
| 12 | — | — | — | — | — | — | — | — | — | — | — | — | 1.00 | 0.83 | 0.71 | 0.64 | 0.64 | 0.63 | 0.61 | 0.60 | 0.58 | 0.60 | 0.60 | 0.58 | 0.56 | 0.57 | 0.56 | 0.55 | 0.57 | 0.54 | 0.52 | 0.58 |
| 13 | — | — | — | — | — | — | — | — | — | — | — | — | — | 1.00 | 0.83 | 0.74 | 0.70 | 0.67 | 0.64 | 0.63 | 0.60 | 0.63 | 0.63 | 0.64 | 0.62 | 0.61 | 0.62 | 0.61 | 0.61 | 0.57 | 0.55 | 0.56 |
| 14 | — | — | — | — | — | — | — | — | — | — | — | — | — | — | 1.00 | 0.81 | 0.74 | 0.71 | 0.68 | 0.66 | 0.63 | 0.65 | 0.64 | 0.64 | 0.62 | 0.61 | 0.60 | 0.59 | 0.58 | 0.54 | 0.54 | 0.59 |
| 15 | — | — | — | — | — | — | — | — | — | — | — | — | — | — | — | 1.00 | 0.88 | 0.80 | 0.75 | 0.70 | 0.66 | 0.65 | 0.63 | 0.62 | 0.59 | 0.57 | 0.57 | 0.56 | 0.55 | 0.53 | 0.55 | 0.59 |
| 16 | — | — | — | — | — | — | — | — | — | — | — | — | — | — | — | — | 1.00 | 0.86 | 0.79 | 0.74 | 0.69 | 0.67 | 0.65 | 0.62 | 0.59 | 0.57 | 0.57 | 0.57 | 0.55 | 0.52 | 0.59 | 0.59 |
| 17 | — | — | — | — | — | — | — | — | — | — | — | — | — | — | — | — | — | 1.00 | 0.90 | 0.84 | 0.78 | 0.74 | 0.72 | 0.69 | 0.66 | 0.63 | 0.61 | 0.60 | 0.61 | 0.58 | 0.66 | 0.65 |
| 18 | — | — | — | — | — | — | — | — | — | — | — | — | — | — | — | — | — | — | 1.00 | 0.91 | 0.85 | 0.80 | 0.77 | 0.73 | 0.70 | 0.68 | 0.66 | 0.64 | 0.64 | 0.61 | 0.66 | 0.66 |
| 19 | — | — | — | — | — | — | — | — | — | — | — | — | — | — | — | — | — | — | — | 1.00 | 0.91 | 0.85 | 0.80 | 0.77 | 0.73 | 0.70 | 0.67 | 0.66 | 0.65 | 0.63 | 0.66 | 0.62 |
| 20 | — | — | — | — | — | — | — | — | — | — | — | — | — | — | — | — | — | — | — | — | 1.00 | 0.91 | 0.86 | 0.82 | 0.77 | 0.74 | 0.71 | 0.69 | 0.68 | 0.65 | 0.71 | 0.69 |
| 21 | — | — | — | — | — | — | — | — | — | — | — | — | — | — | — | — | — | — | — | — | — | 1.00 | 0.93 | 0.88 | 0.84 | 0.81 | 0.78 | 0.76 | 0.75 | 0.71 | 0.75 | 0.67 |
| 22 | — | — | — | — | — | — | — | — | — | — | — | — | — | — | — | — | — | — | — | — | — | — | 1.00 | 0.95 | 0.90 | 0.87 | 0.84 | 0.81 | 0.80 | 0.75 | 0.76 | 0.67 |
| 23 | — | — | — | — | — | — | — | — | — | — | — | — | — | — | — | — | — | — | — | — | — | — | — | 1.00 | 0.94 | 0.89 | 0.86 | 0.83 | 0.82 | 0.76 | 0.77 | 0.70 |
| 24 | — | — | — | — | — | — | — | — | — | — | — | — | — | — | — | — | — | — | — | — | — | — | — | — | 1.00 | 0.92 | 0.88 | 0.85 | 0.83 | 0.77 | 0.78 | 0.68 |
| 25 | — | — | — | — | — | — | — | — | — | — | — | — | — | — | — | — | — | — | — | — | — | — | — | — | — | 1.00 | 0.93 | 0.88 | 0.81 | 0.75 | 0.73 | 0.60 |
| 26 | — | — | — | — | — | — | — | — | — | — | — | — | — | — | — | — | — | — | — | — | — | — | — | — | — | — | 1.00 | 0.92 | 0.80 | 0.73 | 0.71 | 0.62 |
| 27 | — | — | — | — | — | — | — | — | — | — | — | — | — | — | — | — | — | — | — | — | — | — | — | — | — | — | — | 1.00 | 0.90 | 0.81 | 0.74 | 0.70 |
| 28 | — | — | — | — | — | — | — | — | — | — | — | — | — | — | — | — | — | — | — | — | — | — | — | — | — | — | — | — | 1.00 | 0.86 | 0.81 | 0.66 |
| 29 | — | — | — | — | — | — | — | — | — | — | — | — | — | — | — | — | — | — | — | — | — | — | — | — | — | — | — | — | — | 1.00 | 0.87 | 0.72 |
| 30 | — | — | — | — | — | — | — | — | — | — | — | — | — | — | — | — | — | — | — | — | — | — | — | — | — | — | — | — | — | — | 1.00 | 0.78 |
| 31 | — | — | — | — | — | — | — | — | — | — | — | — | — | — | — | — | — | — | — | — | — | — | — | — | — | — | — | — | — | — | — | 1.00 |

Diff-Vector CosSim

| Layer | 0 | 1 | 2 | 3 | 4 | 5 | 6 | 7 | 8 | 9 | 10 | 11 | 12 | 13 | 14 | 15 | 16 | 17 | 18 | 19 | 20 | 21 | 22 | 23 | 24 | 25 | 26 | 27 | 28 | 29 | 30 | 31 |
|---|---|---|---|---|---|---|---|---|---|---|---|---|---|---|---|---|---|---|---|---|---|---|---|---|---|---|---|---|---|---|---|---|
| 0 | 1.00 | 0.17 | 0.13 | 0.08 | 0.05 | 0.03 | 0.03 | 0.03 | 0.03 | 0.01 | 0.01 | 0.01 | 0.03 | 0.05 | 0.05 | 0.05 | 0.05 | 0.05 | 0.04 | 0.04 | 0.04 | 0.03 | 0.03 | 0.04 | 0.03 | 0.03 | 0.02 | 0.01 | -0.00 | 0.00 | 0.10 | 0.02 |
| 1 | — | 1.00 | 0.18 | 0.10 | 0.06 | 0.04 | 0.03 | 0.03 | 0.02 | 0.01 | 0.01 | 0.01 | 0.03 | 0.05 | 0.05 | 0.05 | 0.05 | 0.06 | 0.04 | 0.04 | 0.04 | 0.03 | 0.03 | 0.04 | 0.03 | 0.03 | 0.02 | 0.01 | 0.00 | 0.00 | 0.15 | 0.01 |
| 2 | — | — | 1.00 | 0.20 | 0.11 | 0.07 | 0.06 | 0.05 | 0.04 | 0.02 | 0.02 | 0.02 | 0.03 | 0.05 | 0.05 | 0.05 | 0.05 | 0.04 | 0.05 | 0.04 | 0.04 | 0.04 | 0.05 | 0.04 | 0.03 | 0.03 | 0.02 | 0.00 | 0.00 | 0.00 | 0.13 | 0.01 |
| 3 | — | — | — | 1.00 | 0.19 | 0.10 | 0.07 | 0.06 | 0.05 | 0.03 | 0.02 | 0.02 | 0.04 | 0.05 | 0.05 | 0.06 | 0.06 | 0.05 | 0.05 | 0.04 | 0.05 | 0.03 | 0.03 | 0.04 | 0.03 | 0.02 | 0.01 | 0.00 | 0.00 | 0.00 | 0.10 | 0.02 |
| 4 | — | — | — | — | 1.00 | 0.21 | 0.13 | 0.09 | 0.07 | 0.05 | 0.03 | 0.03 | 0.05 | 0.07 | 0.06 | 0.07 | 0.07 | 0.06 | 0.05 | 0.04 | 0.05 | 0.03 | 0.03 | 0.04 | 0.03 | 0.02 | 0.01 | -0.01 | 0.00 | 0.00 | 0.06 | 0.01 |
| 5 | — | — | — | — | — | 1.00 | 0.21 | 0.15 | 0.09 | 0.07 | 0.06 | 0.04 | 0.05 | 0.07 | 0.06 | 0.07 | 0.07 | 0.07 | 0.06 | 0.05 | 0.05 | 0.04 | 0.05 | 0.04 | 0.04 | 0.02 | 0.01 | 0.00 | 0.00 | 0.00 | 0.11 | 0.02 |
| 6 | — | — | — | — | — | — | 1.00 | 0.25 | 0.12 | 0.09 | 0.08 | 0.04 | 0.06 | 0.07 | 0.08 | 0.08 | 0.07 | 0.06 | 0.07 | 0.05 | 0.06 | 0.05 | 0.05 | 0.03 | 0.03 | 0.02 | 0.00 | 0.01 | 0.00 | 0.06 | 0.02 | |
| 7 | — | — | — | — | — | — | — | 1.00 | 0.24 | 0.17 | 0.12 | 0.08 | 0.09 | 0.10 | 0.08 | 0.09 | 0.09 | 0.09 | 0.08 | 0.07 | 0.06 | 0.04 | 0.05 | 0.05 | 0.06 | 0.05 | 0.04 | 0.02 | 0.02 | 0.01 | 0.05 | 0.02 |
| 8 | — | — | — | — | — | — | — | — | 1.00 | 0.28 | 0.16 | 0.10 | 0.12 | 0.10 | 0.09 | 0.09 | 0.10 | 0.09 | 0.08 | 0.06 | 0.07 | 0.05 | 0.05 | 0.06 | 0.05 | 0.04 | 0.02 | 0.02 | 0.01 | 0.01 | 0.05 | 0.02 |
| 9 | — | — | — | — | — | — | — | — | — | 1.00 | 0.27 | 0.19 | 0.17 | 0.12 | 0.11 | 0.11 | 0.11 | 0.11 | 0.09 | 0.08 | 0.07 | 0.05 | 0.05 | 0.06 | 0.04 | 0.02 | 0.02 | 0.01 | 0.01 | 0.01 | 0.06 | 0.02 |
| 10 | — | — | — | — | — | — | — | — | — | — | 1.00 | 0.21 | 0.17 | 0.12 | 0.12 | 0.12 | 0.11 | 0.10 | 0.09 | 0.07 | 0.07 | 0.05 | 0.06 | 0.05 | 0.05 | 0.03 | 0.02 | 0.00 | 0.01 | 0.02 | 0.07 | 0.02 |
| 11 | — | — | — | — | — | — | — | — | — | — | — | 1.00 | 0.24 | 0.20 | 0.16 | 0.14 | 0.12 | 0.10 | 0.09 | 0.08 | 0.07 | 0.06 | 0.06 | 0.04 | 0.02 | 0.03 | 0.01 | 0.02 | 0.07 | 0.02 | | |
| 12 | — | — | — | — | — | — | — | — | — | — | — | — | 1.00 | 0.25 | 0.21 | 0.17 | 0.15 | 0.12 | 0.10 | 0.09 | 0.09 | 0.07 | 0.06 | 0.07 | 0.06 | 0.05 | 0.03 | 0.03 | 0.01 | 0.02 | 0.07 | 0.02 |
| 13 | — | — | — | — | — | — | — | — | — | — | — | — | — | 1.00 | 0.27 | 0.19 | 0.18 | 0.14 | 0.12 | 0.10 | 0.09 | 0.07 | 0.06 | 0.07 | 0.05 | 0.03 | 0.02 | 0.01 | 0.02 | 0.02 | 0.07 | 0.02 |
| 14 | — | — | — | — | — | — | — | — | — | — | — | — | — | — | 1.00 | 0.29 | 0.22 | 0.15 | 0.13 | 0.12 | 0.12 | 0.09 | 0.08 | 0.08 | 0.07 | 0.06 | 0.04 | 0.03 | 0.02 | 0.02 | 0.08 | 0.02 |
| 15 | — | — | — | — | — | — | — | — | — | — | — | — | — | — | — | 1.00 | 0.28 | 0.21 | 0.17 | 0.15 | 0.14 | 0.12 | 0.10 | 0.10 | 0.09 | 0.07 | 0.05 | 0.04 | 0.03 | 0.03 | 0.10 | 0.02 |
| 16 | — | — | — | — | — | — | — | — | — | — | — | — | — | — | — | — | 1.00 | 0.29 | 0.25 | 0.19 | 0.17 | 0.15 | 0.13 | 0.13 | 0.11 | 0.08 | 0.06 | 0.05 | 0.04 | 0.04 | 0.15 | 0.03 |
| 17 | — | — | — | — | — | — | — | — | — | — | — | — | — | — | — | — | — | 1.00 | 0.35 | 0.28 | 0.25 | 0.22 | 0.19 | 0.20 | 0.16 | 0.13 | 0.11 | 0.09 | 0.08 | 0.07 | 0.18 | 0.04 |
| 18 | — | — | — | — | — | — | — | — | — | — | — | — | — | — | — | — | — | — | 1.00 | 0.36 | 0.31 | 0.28 | 0.22 | 0.23 | 0.21 | 0.14 | 0.10 | 0.10 | 0.09 | 0.07 | 0.18 | 0.04 |
| 19 | — | — | — | — | — | — | — | — | — | — | — | — | — | — | — | — | — | — | — | 1.00 | 0.36 | 0.31 | 0.29 | 0.23 | 0.21 | 0.16 | 0.11 | 0.10 | 0.10 | 0.08 | 0.18 | 0.05 |
| 20 | — | — | — | — | — | — | — | — | — | — | — | — | — | — | — | — | — | — | — | — | 1.00 | 0.44 | 0.33 | 0.32 | 0.31 | 0.19 | 0.15 | 0.12 | 0.13 | 0.08 | 0.20 | 0.05 |
| 21 | — | — | — | — | — | — | — | — | — | — | — | — | — | — | — | — | — | — | — | — | — | 1.00 | 0.46 | 0.43 | 0.41 | 0.24 | 0.19 | 0.17 | 0.17 | 0.09 | 0.20 | 0.05 |
| 22 | — | — | — | — | — | — | — | — | — | — | — | — | — | — | — | — | — | — | — | — | — | — | 1.00 | 0.49 | 0.47 | 0.30 | 0.23 | 0.22 | 0.20 | 0.12 | 0.24 | 0.07 |
| 23 | — | — | — | — | — | — | — | — | — | — | — | — | — | — | — | — | — | — | — | — | — | — | — | 1.00 | 0.57 | 0.34 | 0.28 | 0.23 | 0.21 | 0.12 | 0.24 | 0.06 |
| 24 | — | — | — | — | — | — | — | — | — | — | — | — | — | — | — | — | — | — | — | — | — | — | — | — | 1.00 | 0.38 | 0.32 | 0.29 | 0.25 | 0.12 | 0.25 | 0.06 |
| 25 | — | — | — | — | — | — | — | — | — | — | — | — | — | — | — | — | — | — | — | — | — | — | — | — | — | 1.00 | 0.30 | 0.21 | 0.17 | 0.11 | 0.17 | 0.04 |
| 26 | — | — | — | — | — | — | — | — | — | — | — | — | — | — | — | — | — | — | — | — | — | — | — | — | — | — | 1.00 | 0.30 | 0.18 | 0.10 | 0.17 | 0.05 |
| 27 | — | — | — | — | — | — | — | — | — | — | — | — | — | — | — | — | — | — | — | — | — | — | — | — | — | — | — | 1.00 | 0.26 | 0.15 | 0.20 | 0.10 |
| 28 | — | — | — | — | — | — | — | — | — | — | — | — | — | — | — | — | — | — | — | — | — | — | — | — | — | — | — | — | 1.00 | 0.27 | 0.30 | 0.07 |
| 29 | — | — | — | — | — | — | — | — | — | — | — | — | — | — | — | — | — | — | — | — | — | — | — | — | — | — | — | — | — | 1.00 | 0.30 | 0.06 |
| 30 | — | — | — | — | — | — | — | — | — | — | — | — | — | — | — | — | — | — | — | — | — | — | — | — | — | — | — | — | — | — | 1.00 | 0.22 |
| 31 | — | — | — | — | — | — | — | — | — | — | — | — | — | — | — | — | — | — | — | — | — | — | — | — | — | — | — | — | — | — | — | 1.00 |

# G STEERING VECTOR PERSISTENCE. LOGIT-LENS

In Section 5 we showed that last-layer shifts induced by steering vectors are aligned and decrease the probability of non-English tokens. Figure 15 lists the tokens with the most negative cosine similarity to the last-layer shift for the layer-18 steering vector, which is representative of other layers. Figure 16 gives ChatGPT's language classification of these tokens. Figure 17 presents layer-wise histograms of the cosine-similarity distributions, one for each steering vector at layer $i$.

| | ꙡ | ნ | 暎 | �castle | ∞ | 홓 | 넀 | ʒ | 强 | 圡 | 怡 | ર્ | 沓 | 瀜 | 뗀 | 瀾 |
|---|---|---|---|---|---|---|---|---|---|---|---|---|---|---|---|---|
| Cos. Sim. | -0.63 | -0.62 | -0.62 | -0.62 | -0.62 | -0.62 | -0.62 | -0.62 | -0.62 | -0.62 | -0.62 | -0.62 | -0.62 | -0.62 | -0.62 | -0.62 |
| Dot Prod. | -280.28 | -278.65 | -278.8 | -279.05 | -278.63 | -278.58 | -278.61 | -278.89 | -278.44 | -278.58 | -278.8 | -278.88 | -279.18 | -278.89 | -280.12 | -278.76 |

Figure 15: Tokens with the most negative cosine similarity to the mean shift induced by the layer-18 steering vector (computed against token unembeddings; dot products shown).

Below is a list of tokens. What languages do they originate from and what topic clusters does it contain?

ꀨ ൳ 暎 �castle ꀱ 훓 났 ꒛ 強 ㊏ �店 侭 꺵 澸 똅 灂 �口 琸 ꒚ \OptionsResolver 깻 쥣 烻 뗐 焆 灣 밉 ꒐ 圡 ꕖ ꑂ ꒳ ◆ ꒖ 앒 ꂚ 疅 붉 컨 ꀂ ﻨ ꒜ ꒐ 稑 ﺻ 挼 뇗 ☃ 똣 ꒜

Thought for 28s

Here's what's in that grab-bag of tokens, grouped by script/language and then by "topic" clusters I can infer.

## Languages / scripts represented

- **Korean (Hangul + Jamo):** 훓, 났, �口, 똅, 깻, 쥣, 뗐, 붉, 컨, 앒, 뇗, 똣, 밉, ㊏ (enclosed Hangul).
- **Chinese/Japanese/Korean (CJK Unified Ideographs):** 暎, 焆, 強, 澸, 灂, 琸, 烻, 焆, 灣, 挼, 稑, 恬, 圡 (rare/uncommon Hàn characters used in Chinese; they exist in Japanese/Korean CJK sets too).
- **Buginese (Indonesia, used for Bugis/Makassarese):** ൳, ꒛, ꒐, ꒚, ꒳, ꒐.
- **Tai Tham (Lanna; used for Northern Thai / Tai Lü / Lao Tham):** ꀱ, ꀂ, ꂚ, ꁂ.
- **Tai Viet (used for Tai Dam / Tai Dón):** ꀨ.
- **Javanese (Indonesia):** ꕖ.
- **Arabic script (presentation forms; used for Arabic/Persian/Urdu, etc.):** ꒜, ﺻ, ﻨ (Arabic Presentation Forms), plus ◆ (replacement char often seen when Arabic text is mis-decoded).
- **Canadian Aboriginal Syllabics (e.g., Inuktitut/Cree families):** ꕖ.
- **Ogham (Old/Primitive Irish):** ᚏ.
- **Latin (programming / PHP namespaced identifier):** `\OptionsResolver`.
- **Symbols / compatibility forms:** ☃ (weather symbol "black snowman"), ꒐ (a fullwidth/compatibility form character), ㊏ (enclosed CJK/Hangul).

## Topic clusters you can read off the list

1. **Korean cluster (Hangul syllables & enclosed forms):** 훓, 났, …, 밉, ㊏.
2. **CJK "water & fire" semantics cluster:** many characters built from the water radical 氵 (澸, 灂, 灣, 挼) and fire 灬/火 (焆, 烻, 焆), plus grain (稑) and jade (琸).
3. **Southeast Asian historic/vernacular scripts cluster:** Buginese (൳, ꒛, ꒐, ꒚, ꒳, ꒐), Tai Tham (ꀱ, ꀂ, ꂚ, ꁂ), Tai Viet (ꀨ), Javanese (ꕖ).
4. **Arabic presentation/encoding cluster:** ꒜, ﺻ, ﻨ and the ◆ replacement character (suggesting prior encoding issues).
5. **Indigenous/ancient scripts cluster:** Canadian Aboriginal Syllabics (ꕖ) and Ogham (ᚏ).
6. **Programming / PHP framework cluster:** `\OptionsResolver` (Symfony component naming style).
7. **Symbols & compatibility forms cluster:** ☃ (weather/meteorological symbol), ꒐ (halfwidth/fullwidth block), ㊏ (enclosed CJK/Hangul).

If you want, I can lay these out in a tidy table (token → script → typical languages → notes).

Figure 16: **GPT's classification of the tokens from Figure 15**

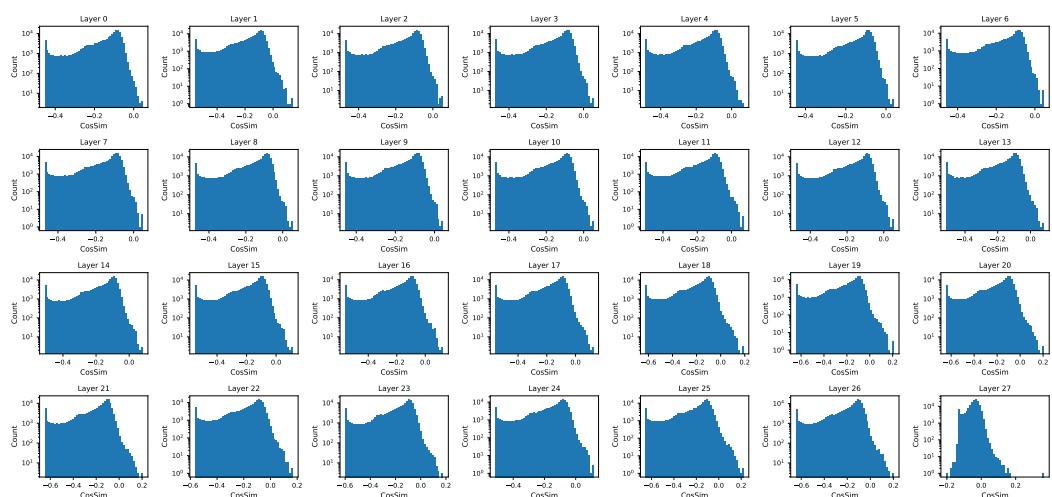

Figure 17: **Layer-wise cosine similarity between the mean shift and token unembeddings.** For each layer $i$, the histograms show $\cos\big(\mathbb{E}_x[\Delta F_{<L,i}(x)], U_t\big)$ over tokens $t$ (log-scale counts). The distributions are largely left-skewed, indicating negative alignment with most tokens and thus a broad reduction in their probabilities.

# H  STEERING VECTOR PERSISTENCE. $\mathbb{E}[\Delta F_{<\ell,i}(x)]$ ALIGNMENT. LLAMA3.1-8B-IT

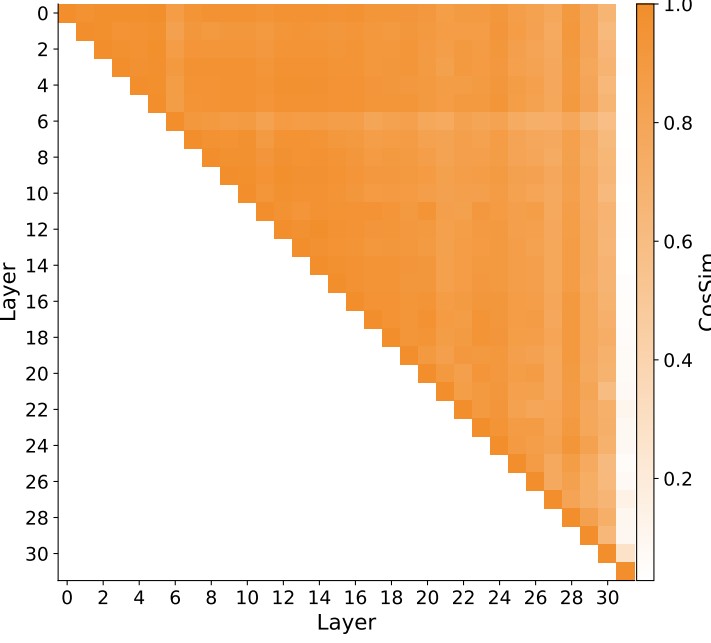

Figure 18: **Similarity of steering-induced unembedding biases. LLaMa3.1-8B-It.** Each cell shows the cosine similarity between the average final-layer shifts $\mathbb{E}[\Delta F_{<L,i}]$ and $\mathbb{E}[\Delta F_{<L,j}]$ induced by steering at layers $i$ and $j$. High similarity across $i, j < L$ indicates a shared effect on the unembedding regardless of where steering is applied. The last-layer shift implements another mechanism. The mean cosine similarity of all pairs $i, j < L$ (up to the last layer) is $0.89$.

## I LAST LAYER. LOGIT LENS

Table 4: **Last Layer – logit-lens.** Cosine similarities and dot-product scores between the last-layer steering vector (trained in isolation) and the unembedding vectors of the top-10 tokens for `Qwen2.5-Math-7B` and `LLaMa3.1-8B-It`.

| Qwen2.5-Math-7B | | | | | | | | | |
|---|---|---|---|---|---|---|---|---|---|
| To | ] | To | So | _to | \ | } | For | .To | -to |
| Cos. Sim. 0.37 | 0.16 | 0.16 | 0.15 | 0.14 | 0.14 | 0.13 | 0.13 | 0.12 | 0.12 |
| Dot Prod. 42.5 | 19.12 | 18.62 | 19.12 | 16.88 | 19.75 | 15.69 | 14.19 | 17.0 | 18.62 |

| LLaMa3.1-8B-It | | | | | | | | | |
|---|---|---|---|---|---|---|---|---|---|
| final | Step | format | Final | final | final | Steps | Final | _final | solution |
| Cos. Sim. 0.12 | 0.11 | 0.09 | 0.09 | 0.08 | 0.08 | 0.08 | 0.08 | 0.08 | 0.08 |
| Dot Prod. 1.69 | 1.32 | 1.17 | 1.09 | 0.71 | 1.01 | 1.02 | 0.93 | 0.83 | 0.95 |

## J LAST LAYER. LLAMA3.1-8B-IT

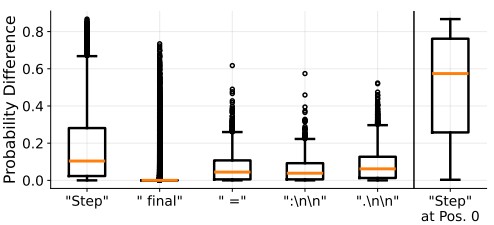
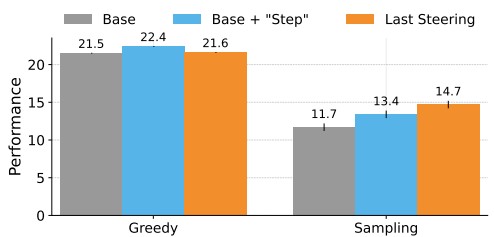

(a) Distribution of the change in token probability ($\Delta P$) produced by the single last-layer steering vector on Qwen-2.5-Math-7B. The box-plots summarise 256 DeepScaleR completions and display (i) the five tokens with the highest maximum $\Delta P$ and (ii) a separate distribution for "*To*" when it appears at the first generated position ("*To*" at Pos. 0).

(b) Prepending "*To*" to every prompt (Base + "*To*") raises performance by 10–11 absolute points under both greedy and sampling decoding by the base model – more than half of the gain achieved by explicit last-layer steering.

Figure 19: **Last Layer Analysis.** The left panel shows how last-layer steering concentrates its impact on starting with token "*To*"; the right panel confirms that this single-token boost translates into a substantial portion of the observed performance improvement.

We analyze the last-layer steering vector for LLaMa3.1-8B-It using the procedure in Section 6. Table 4 (see Appendix I) reports the `logit-lens` scores. Two observations stand out: (i) the vector is only weakly aligned with any single token – the largest cosine similarity is $0.12$ – and (ii) the highest-scoring tokens are variations of "*final*" and "*Step*". Much of the vector's effect is concentrated on "*Step*" at the first generated position (Figure 19a).

Prepending "*Step*" to each prompt improves the peformance of the *base* model under both *Sampling* and *Greedy* decoding. Interestingly, in the *Greedy* setting this prefix even outperforms last-layer steering, plausibly because a last-layer steering vector cannot condition its influence on position and thus perturbs subsequent steps.

## K VALUE STEERING ADDS A LINEAR TERM TO MHA

The following derivation holds when we ignore the pre-attention LayerNorm (LN). While this is a strong assumption – that LN does not alter the steering vector's trajectory – the experiment in

Section 7 shows that a post-attention steering vector attains the same performance as a pre-attention one, indicating that the pre-attention vector indeed does not act through attention.

**Claim.** Let $U \in \mathbb{R}^{T \times d_{\text{model}}}$ and define the (row-wise) attention

$$A(U) = \text{Softmax}\left(\frac{UW_i^Q(UW_i^K)^\top}{\sqrt{d_k}}\right) \in \mathbb{R}^{T \times T}, \quad A(U)\mathbf{1} = \mathbf{1}.$$

For head $i$,

$$H_i(U) = A(U)\,UW_i^V.$$

Let a steering vector $s \in \mathbb{R}^{d_{\text{model}}}$ be added to the *values* of head $i$ for every token, and set $S = \mathbf{1}s^\top \in \mathbb{R}^{T \times d_{\text{model}}}$. Then

$$\begin{aligned}
H_i^{(+s)}(U) &= A(U)\,(U+S)W_i^V \\
&= A(U)UW_i^V + A(U)SW_i^V \\
&= H_i(U) + SW_i^V \qquad (\text{since } A(U)\mathbf{1} = \mathbf{1}).
\end{aligned}$$

Writing $W^O = \begin{bmatrix} W_1^O \\ \cdots \\ W_h^O \end{bmatrix}$ by heads, the multi-head output satisfies

$$\boxed{\text{MHA}(U+S) = \text{MHA}(U) + S\,W_i^V W_i^O}$$

and is independent of the attention pattern.

## L  PENULTIMATE-LAYER STEERING VECTOR IN LLaMa3.1-8B-It

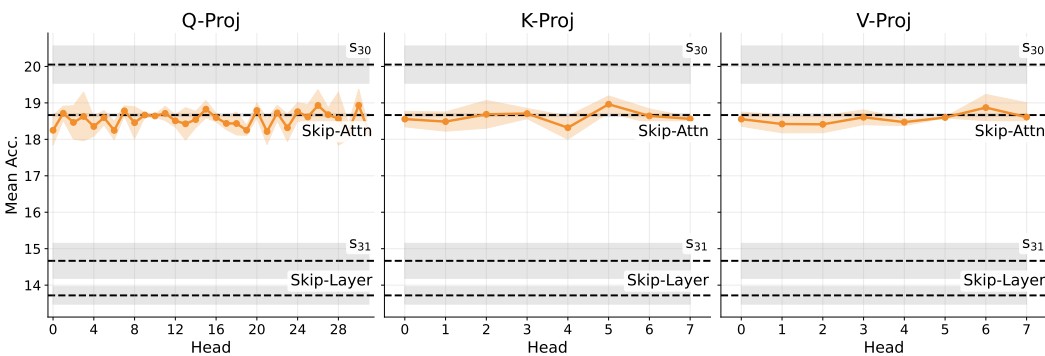

Figure 20: **Penultimate-layer steering in LLaMa3.1-8B-It.** Mean accuracy when the penultimate-layer vector $s_{30}$ is injected into a single $Q$ (left), $K$ (center), or $V$ (right) projection of the final block. Steering any single projection stays near *Skip-Attn* and below $s_{30}$.

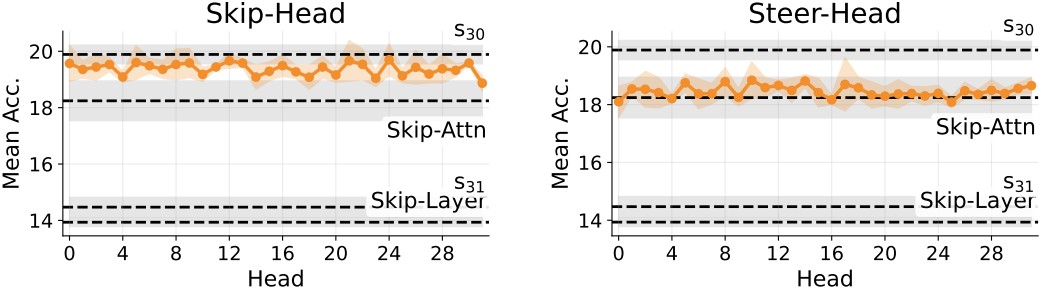

Figure 21: **Penultimate-layer steering in LLaMa3.1-8B-It.** Mean accuracy when applying $s_{30}$ at the final block by patching whole heads: *Skip-Head* (left, steer all except head $i$) and *Steer-Head* (right, steer only head $i$). No single head closes the gap between *Skip-Attn* and $s_{30}$, indicating a cooperative multi-head effect.

Projection-level patching (**Steer–Q/K/V**) did not reveal the source of the gain (Figure 20). We therefore patched entire heads using two setups: **Steer–Head** ($H_i(U) \mapsto H_i(U + s_{l-1})$) and **Skip–Head** (leave $H_i(U)$ unchanged while steering all other heads). In Figure 21, two baselines mirror the Qwen result: *Skip-Layer* performs close to $s_{31}$, indicating a direct unembedding effect, and *Skip-Attn* retains about 70% of the $s_{30}$ gain, suggesting much of the impact bypasses attention. No single head closes the remaining gap between $s_{30}$ and *Skip–Attn*, pointing to a cooperative multi-head mechanism and the importance of attention layer for $s_{30}$'s performance; resolving this is left for future work.

However, training the steering vector in the post-attention residual stream yields performance indistinguishable from $s_{30}$ (mean accuracy $19.9 \pm 0.1$), suggesting either that the vector effectively bypasses attention or that comparable performance can be achieved via the MLP alone.

## M    UNNORMALIZED TRANSFER PERFORMANCE

Table 5: **Transferability of steering vectors within the Qwen2.5 family**. Each cell shows the mean performance change when the steering vector trained for the *Donor* model is applied to the *Recipient* model. The "None" column denotes the non-trained models' performance.

| Family | Recipient | Donor | | | |
|---|---|---|---|---|---|
| | | **None** | **Base** | **Instruct** | **Math** |
| Qwen2.5-1.5B | Base | $0.52 \pm 0.12$ | $22.72 \pm 0.53$ | $9.02 \pm 1.66$ | $7.57 \pm 0.67$ |
| | Instruct | $13.44 \pm 1.17$ | $23.22 \pm 0.35$ | $23.82 \pm 0.28$ | $16.64 \pm 0.27$ |
| | Math | $11.33 \pm 1.09$ | $19.48 \pm 1.18$ | $16.09 \pm 1.48$ | $34.11 \pm 0.28$ |
| Qwen2.5-7B | Base | $12.04 \pm 5.85$ | $36.44 \pm 0.15$ | $20.78 \pm 3.43$ | $30.0 \pm 0.14$ |
| | Instruct | $35.82 \pm 0.14$ | $37.51 \pm 0.44$ | $38.89 \pm 0.27$ | $34.78 \pm 0.07$ |
| | Math | $14.33 \pm 1.75$ | $23.42 \pm 1.76$ | $15.84 \pm 1.96$ | $42.82 \pm 0.25$ |
| LLaMa3.1-8B | Base | $0.91 \pm 0.11$ | $9.18 \pm 0.22$ | $0.95 \pm 0.11$ | — |
| | Instruct | $11.81 \pm 0.43$ | $11.66 \pm 0.46$ | $26.14 \pm 0.43$ | — |

Table 6: **Transferability of steering vectors within the Qwen2.5 family. Generation Length**. Each cell shows the mean generation length change when the steering vector trained for the *Donor* model is applied to the *Recipient* model. The "None" column denotes the non-trained models' performance.

| Family | Recipient | Donor | | | |
|---|---|---|---|---|---|
| | | **None** | **Base** | **Instruct** | **Math** |
| Qwen2.5-1.5B | Base | $3816 \pm 810$ | $1318 \pm 10$ | $2945 \pm 221$ | $3727 \pm 221$ |
| | Instruct | $703 \pm 6$ | $1362 \pm 20$ | $1347 \pm 12$ | $1864 \pm 4$ |
| | Math | $1292 \pm 48$ | $1248 \pm 34$ | $1362 \pm 50$ | $1064 \pm 6$ |
| Qwen2.5-7B | Base | $1153 \pm 36$ | $827 \pm 6$ | $1616 \pm 18$ | $1924 \pm 38$ |
| | Instruct | $793 \pm 4$ | $831 \pm 6$ | $1053 \pm 11$ | $966 \pm 6$ |
| | Math | $1224 \pm 31$ | $1110 \pm 14$ | $936 \pm 4$ | $1287 \pm 27$ |
| LLaMa3.1-8B | Base | $1258 \pm 48$ | $732 \pm 2$ | $1421 \pm 54$ | — |
| | Instruct | $2812 \pm 184$ | $2172 \pm 65$ | $1198 \pm 28$ | — |

# N    PAIR SINGLE RAW

Table 7: **Pairwise composition of steering vectors.** Mean accuracy (%) across six benchmarks when applying two independently trained steering vectors at once: $s_i$ at layer $i$ and $s_j$ at layer $j$.

Qwen2.5-Math-7B

| Layer | 0 | 1 | 2 | 3 | 4 | 5 | 6 | 7 | 8 | 9 | 10 | 11 | 12 | 13 | 14 | 15 | 16 | 17 | 18 | 19 | 20 | 21 | 22 | 23 | 24 | 25 | 26 | 27 |
|---|---|---|---|---|---|---|---|---|---|---|---|---|---|---|---|---|---|---|---|---|---|---|---|---|---|---|---|---|
| 0 | 34.6 | 35.6 | 36.0 | 37.1 | 36.2 | 36.7 | 36.4 | 38.2 | 38.4 | 39.1 | 38.7 | 39.9 | 39.9 | 40.3 | 41.6 | 41.7 | 41.9 | 40.0 | 40.8 | 39.4 | 38.7 | 39.4 | 38.4 | 37.5 | 37.2 | 39.5 | 39.7 | 37.9 |
| 1 | — | 35.6 | 34.9 | 35.1 | 36.0 | 36.1 | 35.2 | 36.7 | 37.4 | 37.8 | 38.3 | 39.6 | 38.5 | 39.3 | 40.4 | 40.8 | 41.0 | 39.2 | 39.5 | 39.4 | 38.9 | 39.0 | 38.6 | 37.2 | 37.4 | 39.9 | 39.0 | 37.1 |
| 2 | — | — | 35.2 | 36.3 | 36.2 | 36.2 | 36.0 | 37.6 | 37.6 | 38.0 | 38.4 | 39.1 | 39.5 | 40.3 | 41.0 | 41.7 | 41.6 | 39.3 | 40.3 | 39.7 | 39.0 | 39.1 | 38.9 | 36.6 | 36.9 | 39.8 | 39.8 | 38.0 |
| 3 | — | — | — | 36.3 | 32.7 | 36.0 | 35.6 | 38.0 | 37.8 | 37.8 | 39.0 | 40.2 | 39.6 | 40.1 | 40.9 | 41.4 | 41.5 | 39.4 | 40.2 | 40.1 | 38.5 | 39.8 | 39.2 | 37.5 | 37.7 | 40.5 | 39.6 | 38.1 |
| 4 | — | — | — | — | 35.4 | 36.4 | 35.4 | 37.4 | 37.0 | 38.1 | 38.0 | 39.6 | 39.2 | 40.2 | 41.3 | 41.6 | 42.3 | 39.7 | 40.2 | 40.0 | 38.6 | 38.9 | 39.1 | 37.6 | 36.6 | 40.6 | 39.5 | 38.0 |
| 5 | — | — | — | — | — | 37.0 | 35.8 | 37.2 | 37.4 | 38.4 | 38.5 | 39.6 | 40.1 | 40.0 | 41.1 | 41.4 | 41.5 | 39.6 | 41.1 | 40.8 | 39.8 | 39.5 | 39.3 | 38.3 | 37.4 | 40.1 | 39.7 | 37.9 |
| 6 | — | — | — | — | — | — | 36.3 | 36.7 | 36.4 | 37.7 | 37.8 | 39.4 | 38.8 | 39.4 | 41.0 | 41.4 | 41.5 | 39.8 | 40.3 | 39.7 | 39.0 | 39.6 | 38.9 | 37.5 | 36.9 | 40.5 | 39.9 | 37.8 |
| 7 | — | — | — | — | — | — | — | 37.1 | 36.0 | 37.1 | 38.1 | 39.0 | 39.5 | 39.8 | 41.7 | 41.6 | 41.9 | 40.3 | 40.2 | 41.1 | 40.1 | 40.2 | 39.4 | 38.3 | 37.7 | 41.1 | 40.3 | 38.5 |
| 8 | — | — | — | — | — | — | — | — | 36.9 | 35.9 | 37.3 | 38.8 | 38.2 | 39.4 | 40.8 | 40.3 | 41.5 | 39.9 | 40.0 | 40.7 | 40.0 | 40.4 | 39.0 | 38.0 | 38.4 | 40.1 | 41.0 | 38.3 |
| 9 | — | — | — | — | — | — | — | — | — | 37.4 | 37.4 | 38.5 | 39.3 | 39.5 | 39.7 | 40.7 | 40.9 | 39.1 | 39.2 | 40.2 | 40.3 | 40.4 | 40.0 | 38.8 | 38.2 | 40.8 | 40.6 | 39.5 |
| 10 | — | — | — | — | — | — | — | — | — | — | 37.2 | 37.1 | 38.6 | 39.6 | 40.3 | 40.6 | 40.8 | 40.2 | 41.1 | 40.2 | 40.5 | 40.9 | 39.6 | 37.7 | 37.6 | 41.0 | 41.2 | 39.6 |
| 11 | — | — | — | — | — | — | — | — | — | — | — | 37.4 | 38.0 | 38.9 | 39.3 | 40.6 | 41.6 | 39.3 | 40.2 | 40.7 | 40.2 | 40.8 | 39.8 | 38.3 | 37.5 | 40.9 | 41.4 | 39.9 |
| 12 | — | — | — | — | — | — | — | — | — | — | — | — | 37.2 | 37.3 | 39.4 | 40.5 | 41.1 | 39.4 | 40.4 | 40.6 | 40.1 | 40.8 | 40.2 | 39.1 | 37.9 | 41.9 | 41.1 | 39.8 |
| 13 | — | — | — | — | — | — | — | — | — | — | — | — | — | 38.6 | 39.1 | 40.1 | 41.2 | 39.7 | 40.4 | 40.9 | 40.2 | 40.7 | 39.8 | 38.3 | 38.1 | 41.3 | 40.0 | 40.5 |
| 14 | — | — | — | — | — | — | — | — | — | — | — | — | — | — | 39.9 | 39.3 | 40.4 | 39.5 | 40.4 | 41.6 | 41.6 | 41.2 | 40.2 | 39.8 | 39.1 | 43.1 | 42.2 | 41.6 |
| 15 | — | — | — | — | — | — | — | — | — | — | — | — | — | — | — | 39.2 | 39.8 | 38.5 | 39.5 | 40.7 | 41.0 | 41.0 | 39.9 | 39.6 | 39.1 | 42.5 | 40.0 | 42.2 |
| 16 | — | — | — | — | — | — | — | — | — | — | — | — | — | — | — | — | 40.0 | 35.5 | 37.6 | 40.7 | 40.7 | 40.8 | 40.2 | 39.5 | 38.8 | 42.9 | 39.9 | 41.9 |
| 17 | — | — | — | — | — | — | — | — | — | — | — | — | — | — | — | — | — | 36.6 | 32.7 | 39.5 | 39.8 | 39.7 | 39.4 | 37.5 | 37.3 | 41.2 | 38.6 | 40.1 |
| 18 | — | — | — | — | — | — | — | — | — | — | — | — | — | — | — | — | — | — | 37.5 | 36.9 | 37.9 | 38.0 | 38.0 | 34.8 | 36.9 | 39.8 | 34.7 | 40.1 |
| 19 | — | — | — | — | — | — | — | — | — | — | — | — | — | — | — | — | — | — | — | 38.2 | 31.1 | 34.6 | 35.6 | 26.1 | 35.4 | 39.1 | 25.5 | 39.5 |
| 20 | — | — | — | — | — | — | — | — | — | — | — | — | — | — | — | — | — | — | — | — | 36.3 | 33.4 | 31.6 | 18.3 | 33.9 | 38.1 | 18.3 | 37.9 |
| 21 | — | — | — | — | — | — | — | — | — | — | — | — | — | — | — | — | — | — | — | — | — | 37.4 | 32.6 | 34.3 | 35.3 | 36.8 | 24.0 | 38.3 |
| 22 | — | — | — | — | — | — | — | — | — | — | — | — | — | — | — | — | — | — | — | — | — | — | 36.5 | 13.8 | 24.5 | 27.0 | 17.4 | 32.2 |
| 23 | — | — | — | — | — | — | — | — | — | — | — | — | — | — | — | — | — | — | — | — | — | — | — | 23.2 | 15.5 | 35.6 | 12.1 | 33.7 |
| 24 | — | — | — | — | — | — | — | — | — | — | — | — | — | — | — | — | — | — | — | — | — | — | — | — | 24.6 | 34.2 | 12.9 | 34.2 |
| 25 | — | — | — | — | — | — | — | — | — | — | — | — | — | — | — | — | — | — | — | — | — | — | — | — | — | 38.8 | 20.7 | 35.3 |
| 26 | — | — | — | — | — | — | — | — | — | — | — | — | — | — | — | — | — | — | — | — | — | — | — | — | — | — | 37.5 | 36.4 |
| 27 | — | — | — | — | — | — | — | — | — | — | — | — | — | — | — | — | — | — | — | — | — | — | — | — | — | — | — | 30.6 |

LLaMa3.1-8B-It

| Layer | 0 | 1 | 2 | 3 | 4 | 5 | 6 | 7 | 8 | 9 | 10 | 11 | 12 | 13 | 14 | 15 | 16 | 17 | 18 | 19 | 20 | 21 | 22 | 23 | 24 | 25 | 26 | 27 | 28 | 29 | 30 | 31 |
|---|---|---|---|---|---|---|---|---|---|---|---|---|---|---|---|---|---|---|---|---|---|---|---|---|---|---|---|---|---|---|---|---|
| 0 | 21.1 | 17.3 | 17.4 | 17.7 | 18.1 | 19.2 | 19.1 | 19.5 | 21.2 | 22.2 | 20.5 | 22.6 | 22.5 | 20.6 | 21.9 | 21.1 | 22.0 | 20.1 | 19.1 | 22.1 | 19.7 | 20.1 | 22.0 | 20.3 | 20.6 | 19.5 | 19.8 | 20.2 | 19.8 | 20.0 | 20.5 | 20.6 |
| 1 | — | 21.9 | 17.1 | 17.9 | 17.5 | 17.9 | 18.4 | 19.9 | 20.4 | 22.7 | 20.5 | 22.7 | 22.4 | 20.9 | 21.9 | 20.7 | 21.0 | 20.2 | 20.7 | 21.2 | 20.9 | 19.3 | 21.8 | 20.9 | 21.1 | 19.6 | 20.2 | 21.2 | 20.1 | 19.7 | 20.7 | 20.9 |
| 2 | — | — | 22.8 | 18.3 | 18.6 | 18.7 | 19.3 | 19.2 | 20.6 | 22.8 | 20.6 | 22.9 | 20.7 | 21.8 | 21.1 | 21.7 | 20.9 | 20.6 | 22.0 | 21.4 | 21.2 | 22.5 | 21.4 | 22.2 | 20.6 | 21.2 | 21.7 | 21.1 | 20.7 | 22.0 | 21.6 | |
| 3 | — | — | — | 22.6 | 18.5 | 19.4 | 19.5 | 19.4 | 20.8 | 23.8 | 21.6 | 23.5 | 23.2 | 20.8 | 22.8 | 21.7 | 22.6 | 20.8 | 20.8 | 22.6 | 21.4 | 21.4 | 22.5 | 22.0 | 21.8 | 19.6 | 21.1 | 21.4 | 21.0 | 20.1 | 22.4 | 21.6 |
| 4 | — | — | — | — | 21.7 | 18.1 | 19.0 | 19.4 | 19.6 | 22.9 | 20.9 | 22.2 | 22.6 | 20.6 | 22.4 | 20.9 | 21.0 | 20.1 | 20.1 | 20.8 | 21.3 | 20.2 | 21.6 | 20.9 | 21.6 | 12.8 | 18.2 | 20.6 | 20.8 | 11.3 | 21.0 | 21.5 |
| 5 | — | — | — | — | — | 22.3 | 18.8 | 19.0 | 20.7 | 22.1 | 20.7 | 22.3 | 22.9 | 20.5 | 21.8 | 20.5 | 21.6 | 20.8 | 21.5 | 22.4 | 21.4 | 21.2 | 22.7 | 21.5 | 22.0 | 19.3 | 20.7 | 21.1 | 21.0 | 20.7 | 21.9 | 21.3 |
| 6 | — | — | — | — | — | — | 21.8 | 18.5 | 19.9 | 21.5 | 19.4 | 21.9 | 22.1 | 20.6 | 20.7 | 20.4 | 21.6 | 20.9 | 21.4 | 20.8 | 21.7 | 20.9 | 22.3 | 21.5 | 21.7 | 19.4 | 20.6 | 20.8 | 20.8 | 20.7 | 21.5 | 20.7 |
| 7 | — | — | — | — | — | — | — | 22.4 | 18.9 | 19.4 | 19.4 | 18.6 | 21.0 | 21.6 | 20.0 | 20.9 | 20.4 | 21.6 | 20.4 | 20.6 | 20.9 | 21.0 | 20.3 | 22.4 | 20.8 | 21.6 | 14.5 | 19.1 | 21.0 | 20.1 | 18.3 | 21.5 | 20.7 |
| 8 | — | — | — | — | — | — | — | — | 23.0 | 20.9 | 18.6 | 21.1 | 22.4 | 20.9 | 23.0 | 21.6 | 23.3 | 22.4 | 22.7 | 23.2 | 23.2 | 23.7 | 24.2 | 23.6 | 21.9 | 22.4 | 23.5 | 24.0 | 23.8 | 23.8 | 23.1 |
| 9 | — | — | — | — | — | — | — | — | — | 24.7 | 18.7 | 20.4 | 23.6 | 21.4 | 22.9 | 22.7 | 24.4 | 23.0 | 22.7 | 23.7 | 24.0 | 23.7 | 24.9 | 24.0 | 24.7 | 23.3 | 23.4 | 23.7 | 24.3 | 23.7 | 24.4 | 23.8 |
| 10 | — | — | — | — | — | — | — | — | — | — | 23.4 | 17.6 | 20.3 | 19.3 | 20.9 | 20.5 | 23.1 | 21.7 | 21.8 | 22.5 | 22.6 | 23.6 | 22.5 | 23.0 | 21.3 | 20.8 | 22.0 | 22.2 | 22.4 | 22.9 | 22.4 |
| 11 | — | — | — | — | — | — | — | — | — | — | — | 24.2 | 20.9 | 20.8 | 21.7 | 21.6 | 23.7 | 22.7 | 23.1 | 23.8 | 24.3 | 23.3 | 24.1 | 24.0 | 24.2 | 22.4 | 23.3 | 24.0 | 24.1 | 24.5 | 24.1 | 24.0 |
| 12 | — | — | — | — | — | — | — | — | — | — | — | — | 24.5 | 20.8 | 22.5 | 21.7 | 23.4 | 22.7 | 23.3 | 23.8 | 24.2 | 23.5 | 24.3 | 23.9 | 23.7 | 23.2 | 23.5 | 23.6 | 23.3 | 23.1 | 24.1 | 23.0 |
| 13 | — | — | — | — | — | — | — | — | — | — | — | — | — | 23.1 | 20.0 | 19.5 | 20.9 | 20.0 | 20.6 | 20.3 | 21.1 | 20.7 | 20.7 | 21.3 | 21.6 | 19.6 | 20.6 | 21.5 | 21.0 | 20.6 | 21.7 | 21.9 |
| 14 | — | — | — | — | — | — | — | — | — | — | — | — | — | — | 24.2 | 19.7 | 22.0 | 21.3 | 23.4 | 22.9 | 24.3 | 23.4 | 23.4 | 23.6 | 21.5 | 23.0 | 23.5 | 23.3 | 22.2 | 23.5 | 23.4 |
| 15 | — | — | — | — | — | — | — | — | — | — | — | — | — | — | — | 23.5 | 21.1 | 21.3 | 22.1 | 21.0 | 23.0 | 22.3 | 23.0 | 22.6 | 23.1 | 21.1 | 22.5 | 22.7 | 23.0 | 22.0 | 22.5 | 23.1 |
| 16 | — | — | — | — | — | — | — | — | — | — | — | — | — | — | — | — | 22.9 | 20.9 | 21.2 | 21.3 | 21.9 | 20.9 | 22.0 | 22.0 | 21.8 | 20.7 | 21.0 | 22.2 | 22.4 | 20.2 | 22.0 | 22.0 |
| 17 | — | — | — | — | — | — | — | — | — | — | — | — | — | — | — | — | — | 21.9 | 21.0 | 21.2 | 21.3 | 20.2 | 22.4 | 20.9 | 20.7 | 20.0 | 20.7 | 21.6 | 20.9 | 20.9 | 20.6 | 21.0 |
| 18 | — | — | — | — | — | — | — | — | — | — | — | — | — | — | — | — | — | — | 21.5 | 20.2 | 20.7 | 20.7 | 21.9 | 20.4 | 20.5 | 19.4 | 20.6 | 20.4 | 19.5 | 20.4 | 19.9 | 20.6 |
| 19 | — | — | — | — | — | — | — | — | — | — | — | — | — | — | — | — | — | — | — | 22.8 | 21.0 | 21.0 | 21.0 | 21.6 | 20.8 | 19.7 | 21.1 | 21.2 | 20.5 | 20.4 | 21.3 | 22.2 |
| 20 | — | — | — | — | — | — | — | — | — | — | — | — | — | — | — | — | — | — | — | — | 21.9 | 20.5 | 21.3 | 20.7 | 20.5 | 19.6 | 20.9 | 21.9 | 20.6 | 20.4 | 20.9 | 20.8 |
| 21 | — | — | — | — | — | — | — | — | — | — | — | — | — | — | — | — | — | — | — | — | — | 21.1 | 21.7 | 19.7 | 19.5 | 19.4 | 20.4 | 20.8 | 19.1 | 19.6 | 20.0 | 20.5 |
| 22 | — | — | — | — | — | — | — | — | — | — | — | — | — | — | — | — | — | — | — | — | — | — | 22.3 | 21.9 | 21.8 | 20.3 | 21.3 | 22.4 | 21.2 | 21.1 | 21.2 | 21.9 |
| 23 | — | — | — | — | — | — | — | — | — | — | — | — | — | — | — | — | — | — | — | — | — | — | — | 21.1 | 20.0 | 19.3 | 19.6 | 20.9 | 19.1 | 16.6 | 20.1 | 20.1 |
| 24 | — | — | — | — | — | — | — | — | — | — | — | — | — | — | — | — | — | — | — | — | — | — | — | — | 21.4 | 19.0 | 20.4 | 18.8 | 17.2 | 20.2 | 20.6 |
| 25 | — | — | — | — | — | — | — | — | — | — | — | — | — | — | — | — | — | — | — | — | — | — | — | — | — | 21.9 | 13.7 | 18.6 | 19.2 | 16.6 | 20.0 | 21.4 |
| 26 | — | — | — | — | — | — | — | — | — | — | — | — | — | — | — | — | — | — | — | — | — | — | — | — | — | — | 21.7 | 18.8 | 19.6 | 16.6 | 20.0 | 20.7 |
| 27 | — | — | — | — | — | — | — | — | — | — | — | — | — | — | — | — | — | — | — | — | — | — | — | — | — | — | — | 21.6 | 19.5 | 15.3 | 20.2 | 20.0 |
| 28 | — | — | — | — | — | — | — | — | — | — | — | — | — | — | — | — | — | — | — | — | — | — | — | — | — | — | — | — | 20.9 | 13.2 | 19.2 | 20.8 |
| 29 | — | — | — | — | — | — | — | — | — | — | — | — | — | — | — | — | — | — | — | — | — | — | — | — | — | — | — | — | — | 21.1 | 12.3 | 20.3 |
| 30 | — | — | — | — | — | — | — | — | — | — | — | — | — | — | — | — | — | — | — | — | — | — | — | — | — | — | — | — | — | — | 20.0 | 18.5 |
| 31 | — | — | — | — | — | — | — | — | — | — | — | — | — | — | — | — | — | — | — | — | — | — | — | — | — | — | — | — | — | — | — | 14.8 |

# O    CHAT TEMPLATES

Following Liu et al. (2025b), we used two chat templates. For models that support special chat-template tokens, we adopted the *Qwen-Math* template; special tokens for `Qwen2.5-Math-7B` are shown as a representative example. For `LLaMa3.1-8B` – which does not include pretrained special chat-template tokens – we used the *R1* template.

> **Chat Template – Qwen-Math**
>
> ```
> <|im_start|>system Please reason step by step, and put your
> final answer within \boxed{}.<|im_end|> <|im_start|>user
> TASK<|im_end|> <|im_start|>assistant
> ```

```
Chat Template – R1

A conversation between User and Assistant.  The User asks
a question, and the Assistant solves it.  The Assistant
first thinks about the reasoning process in the mind and
then provides the User with the answer.  The reasoning
process is enclosed within <think> </think> and answer is
enclosed within <answer> </answer> tags, respectively, i.e.,
<think> reasoning process here </think> <answer> answer here
</answer>.  \nUser:  TASK\nAssistant:  <think>
```

## P  LLM USE

We used the latest ChatGPT to check grammar and wording. We also asked it to identify the language of non-English tokens and give brief meanings (Figure 16).

## Q  FEATURES WITH TOP CAS

```
plane . ### 7 . Find the distance between (-2,4) and (3,-8). Here , \((x_1, y_1) = (-2,
4)\)) and \((x_2, y_2) = (3, -8)\). Substitute these values into the distance formula : \[ d
= \sqrt{(3 - (-2))^2 + (-8 - 4)^2} \] Simplify inside the parentheses : \[ d = \sqrt{(3 +
2)^2 + (-8 - 4)^2} \] \[ d = \sqrt{5^2 + (-12)^2} \] Calculate the squares : \[ d =
\sqrt{25 + 144} \] Add the results : \[ d = \sqrt{169} \] Take the square root : \[ d =
13 \] So , the distance is \(\boxed{13}\). ### 8 . What is the distance between (-5,1) and
(11,-3)? Here , \((x_1, y_1) = (-5, 1)\) and \((x_2, y_2) = (11, -3)\). Substitute
these values into the distance formula : \[ d = \sqrt{(11 - (-5))^2 + (-3 - 1)^2} \]

( ST = 3 \)). 4 . Now , we need to calculate the distance between point \( P \) at \( (-4, 0)
\) and point \( T \) at \( (8, 5) \). The distance \( d \) between two points \((x_1,
y_1)\) and \((x_2, y_2)\) is given by : \[ d = \sqrt{(x_2 - x_1)^2 + (y_2 - y_1)^2}
\] Let 's calculate this distance using the coordinates of \( P \) and \( T \). ```python import
math # Coordinates of P and T  P = (-4, 0) T = (8, 5) # Calculate the distance between P and
T  distance_PT = math.sqrt((T[0] - P[0])**2 + (T[1] - P[1])**2) print(distance_PT) ```
```output 13.0 ``` The distance from \( P \) to \( T \) is \( 13 \). So the final answer
is : \[ \boxed{13} \]

( ST = 3 \)). 4 . Now , we need to calculate the distance between point \( P \) at \( (-4, 0)
\) and point \( T \) at \( (8, 5) \). The distance \( d \) between two points \((x_1,
y_1)\) and \((x_2, y_2)\) is given by : \[ d = \sqrt{(x_2 - x_1)^2 + (y_2 - y_1)^2}
\] Let 's calculate this distance using the coordinates of \( P \) and \( T \). ```python import
math # Coordinates of P and T  P = (-4, 0) T = (8, 5) # Calculate the distance between P and
T  distance_PT = math.sqrt((T[0] - P[0])**2 + (T[1] - P[1])**2) print(distance_PT) ```
```output 13.0 ``` The distance from \( P \) to \( T \) is \( 13 \). So the final answer
is : \[ \boxed{13} \]
```

Figure 22: $(\ell = 16, \ell + k = 17)$, F-3782, Top-1 in Correctness. This feature seems to be related to "boxed" token.

pairs of books I can choose from my eleven books , we can use the concept of combinations . The formula for combinations ( often den oted as \( C ( n , k ) \) or \( n \choose k \) is given by : \[ C ( n , k ) = \ frac {n ! }{ k ! ( n – k )! } \ \] where : – \( n \) is the total number of items , – \( k \) ) is the number of items to choose . In this case , \( n = 1 1 \) and \( k = 2 \) . Let 's calculate this using Python . ``` python import math # Number of books n = 1 1 # Number of books to choose k = 2 # Calculate the number of combinations num _comb inations = math .com b ( n , k ) print (num _comb inations ) ``` ``` output 5 5 ``` The number of different pairs of books I can choose from my eleven books is \ (\ boxed { 5 5 }\ ).

contains 8 squares . The second ring ( at a distance of 2 units ) contains 1 6 squares . In general , the \( n ^{ th }\) ring ( at a distance of \( n \) units ) contains \( 8 n \) squares . Thus , the number of squares in the \( 5 0 ^{ th }\) ring is \( 8 \ times 5 0 = 4 0 0 \). Let 's write a simple Python code to verify this pattern and calculate the number of unit squares in the \( 5 0 ^{ th }\) ring . ``` python def number _of _s quares _in _nth _ring ( n ): return 8 * n # Calculate the number of unit squares in the 5 0 th ring nth _ring = 5 0 number _of _s quares = number _of _s quares _in _nth _ring ( n th _ring ) print (number _of _s quares ) ``` ``` output 4 0 0 ``` The number of unit squares in the \( 5 0 ^{ th }\) ring is \ (\ boxed { 4 0 0 }\ ).

\ impl ies f \ left (\ frac { 1 }{ 2 }\ right ) = 1 .\ ] Sub stit uting \( f \ left (\ frac { 1 }{ 2 }\ right ) = 1 \) into the first equation , we get : \ [ f ( 2 ) + 1 = 1 \ impl ies f ( 2 ) = 0 .\ ] Let 's verify this solution using Python and symp y to ensure accuracy . ``` python import symp y as sp # Define the symbols f _ 2 , f _half = sp .s ymbols (' f _ 2 f _half ') # Define the equations based on the functional equation eq 1 = sp .E q ( f _ 2 + f _half , 1 ) eq 2 = sp .E q ( f _half – f _ 2 , 1 ) # Solve the system of equations solution = sp .solve (( eq 1 , eq 2 ) , ( f _ 2 , f _half )) # Extract the value of f ( 2 ) f _ 2 _value = solution [ f _ 2 ] print (f _ 2 _value ) ``` ``` output 0 ``` The value of the function \( f ( x )\) at \( x = 2 \) is \ (\ boxed { 0 }\ ).

Figure 23: ($\ell = 16, \ell + k = 20$), F-2515, Top-1 in Correctness. This feature seems to be related to "boxed" token.

\ frac { 1 }{ 2 } + \ frac { 1 }{ 4 } + \ frac { 1 }{ 8 } + \ cd ots \ ] The series on the right is a geometric series with the first term \( a = 1 \) and common ratio \( r = \ frac { 1 }{ 2 }\). The sum of an infinite geometric series is given by \( S = \ frac { a }{ 1 – r }\): \ [ S = \ frac { 1 }{ 1 – \ frac { 1 }{ 2 }} = \ frac { 1 }{ \ frac { 1 }{ 2 }} = 2 \ ] Thus , the sum of the series \ (\ sum _{ n = 1 } ^\ in fty \ frac {n }{ 2 ^n }\ ) is \( 2 \). 5 . ** If you cut a square into four right – angled triangles , what is the area of the largest triangle ? ** If we start with a square of side length \ ( a \ ), cutting it into four right – angled triangles will result in each triangle having legs of length \ (\ frac { a }{ 2 }\). The area of each triangle is : \ [ \ text { Area } = \ frac { 1 }{ 2 } \ times

Hence , $ b _n $ is not an integer . Also , since $ a _n $ is a perfect square , we have $ 2 a _n ( a _n + 1 )$ is a perfect square . Hence , $ b _n $ is not an integer . Therefore , $ b _n $ is not an integer for all $ n \ ge q 0 $. Thus , $ a _n ^ 2 < b _n ^ 2 < ( a _n + 1 )^ 2 $ for all $ n \ ge q 0 $. Hence , $ a _n $ is a perfect square for all $ n \ ge q 0 $. $\ bullet $ $ 2 0 1 7 $ $\ bullet $ But since $ b _n $ is not an integer , we have $ a _n ^ 2 < b _n ^ 2 < ( a _n + 1 )^ 2 $. Thus , $ a _n $ is a perfect square for all $ n \ ge q 0 $. Therefore , $ a _n $ is a perfect square for all [] that $ b _n $ is not an integer , we have $ a ^ 2 < b ^ 2 < ^ 2 $. Since $ a $ is a perfect square , we have $ ^ 2 $. Hence , $ b $ is not an integer . Also , since $ a $ is a perfect square , we have $ a ( a + 1 )...

the item that broke is a second –class item ( since this will leave more first –class items ). So , truck A now has 2 first –class items and 1 second –class item . 2 . Truck B originally carried 4 first –class items and 2 second –class items . After one item broke , it either lost a first –class item or a second –class item . Again , we will consider the worst –case scenario where the item that broke is a second –class item . So , truck B now has 4 first –class items and 1 second –class item . 3 . The remaining total number of first –class items is \( 2 + 4 = 6 \). 4 . The remaining total number of items is \( 8 \). 5 . The probability of selecting a first –class item from the remaining \ ( 8 \) items is the ratio of first –class items to the total number of remaining items . Let 's use

Figure 24: ($\ell = 16, \ell + k = 20$), F-2625, Top-1 in Incorrectness. This feature seems to be related to repeated tokens in different contexts.

# R  DIFFSAE

While vanilla SAE is trained on hidden states directly, DiffSAE trains on the differece of the activation of different models, however in our case difference came for the pathcing hidden states with steering on the previous layer. Define $\boldsymbol{d}^{(\ell)} := \boldsymbol{h}_s^{(\ell)} - \boldsymbol{h}_b^{(\ell)}$, then

$$\boldsymbol{z}^{(\ell)} = \sigma\big(\boldsymbol{W}_{\text{enc}}^{(\ell)} \boldsymbol{h}^{(\ell)} + \boldsymbol{b}_{\text{enc}}^{(\ell)}\big) \in \mathbb{R}^F,$$
$$\widehat{\boldsymbol{d}^{(\ell)}} = \boldsymbol{W}_{\text{dec}}^{(\ell)} \boldsymbol{z}^{(\ell)} + \boldsymbol{b}_{\text{dec}}^{(\ell)},$$

where $\boldsymbol{W}_{\text{enc}}^{\ell} \in \mathbb{R}^{F \times d}$, $\boldsymbol{b}_{\text{enc}}^{(\ell)} \in \mathbb{R}^F$, $\boldsymbol{W}_{\text{dec}}^{\ell} \in \mathbb{R}^{d \times F}$, $\boldsymbol{b}_{\text{dec}}^{(\ell)} \in \mathbb{R}^d$, and $\sigma(\cdot)$ is a sparsity-enforcing function (e.g., BatchTopK (Bussmann et al., 2024)).

# S  ADAPTIVE STEERING. EXAMPLES

Figure 25: **Adaptive steering at layer 5.** Colors indicate the sign and magnitude of the token-wise scaling factor. The layer-5 adaptive vector activates on variables and math-operator tokens.

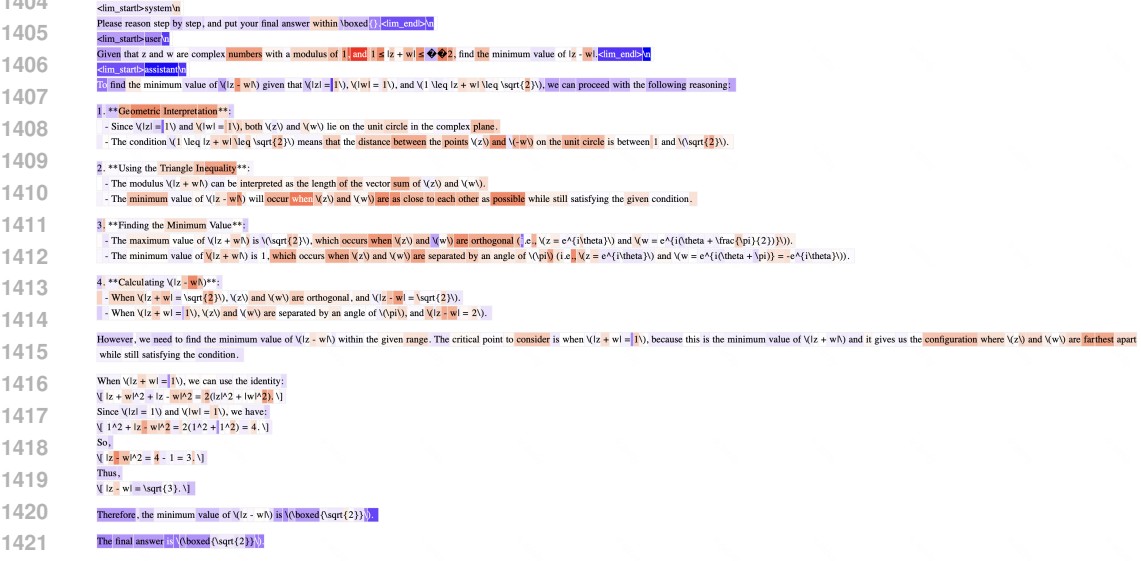

Figure 26: **Adaptive steering at layer 15.** Colors show the sign and magnitude of the token-wise scaling factor. The layer-15 adaptive vector gives positive weight to natural-language reasoning and definition tokens, and negative weight to the instruction prompt and answer-generation tokens.

