# OpenReview forum: "Small Vectors, Big Effects: A Mechanistic Study of RL-Induced Reasoning via Steering Vectors"
_ICLR.cc/2026/Conference — Submitted to ICLR 2026_

### Official Review · Reviewer_TeJy · 2025-10-31

**Soundness:** 2
**Presentation:** 2
**Contribution:** 2
**Rating:** 2
**Confidence:** 3

**Summary:**

This paper proposes to understand the effect of LLM reasoning training by examining steering vectors.
The per-layer steering vectors are obtained by optimizing a standard RLVR objective.

When using a steering vector at a single layer and keeping other layers frozen, the paper finds that:
- All but 2 layers lead to improved accuracy.
- All pre-final layers downweight non-English tokens, while the last layer mostly boosts the probability of outputting opening tokens such as "To" or "Step".

The paper also reports the cosine similarity between the steering vectors, and the changes they incur on representations. The paper considers two quantities per-layer:
- Diff-Diff CosSim, i.e. the similarity between 1) the per-sample change in representation and 2) the averaged change in representation.
- Diff-Vector CosSim, i.e. the similarity between 1) the per-sample change in representation and 2) he steering vector.

While Qwen and Llama models show differences, the overall trend is that:
- Diff-Diff CosSim is generally high, suggesting that the effect of steering vector propagates through layers.
- Diff-Vector CosSim is low.

The paper also shows that steering vectors have some extent of transferability across models.

**Strengths:**

- The paper considers thorough setups, including multiple ways to perform steering, and different ways to interpret the steering effects.
- The paper experiments with transferring steering vectors across models in the same family (i.e. across models within Qwen-1.5B, Qwen-7B, LLaMA-8B, respectively). The extent of success vary across families, which is an interesting insight.

**Weaknesses:**

**Lack of actionable items**: While the paper provides a list of observations, I'm not sure what these observations entail.
- The paper would be stronger if it suggests actionable items, such as showing that the insights from these steering vectors can improve the training of the model.
- The paper could benefit from restructuring; currently it feels like a set of scattered observations.

**Insufficient empirical evidence for some results**: Some claims are based on limited empirical evidence.
- I'm not convinced of the claim of "pre-final layers downweight non-English tokens, since the empirical results are not sufficient.
  - Fig 15 only shows 16 examples which is too small; these could simply be uncommon characters, rather than non-English.
  - In Fig 16, ChatGPT is prompted to distinguish languages.
  - Fig 17 further suggests that a lot of tokens have a strong negative correlation, making the examples in Fig 15 and 16 non-representative.
  - Moreover, there are no comparison of tokens downweighted by the final layer.
- For the SAE feature analysis, Fig 8 only shows limited features, and Fig 7 doesn't say what the "top featuress" are.

**Questions:**

- For the final layer, how do tokens like "To", " To", or "Step" relate to general attention sink tokens?
- For transferring across models, what does higher or lower transfer indicate about the model family? How can we use this knowledge to improve the training recipe?
- Could you highlight key insights or actionable items?

---

> ### Author Response · Authors · 2025-11-21
> **non-English tokens**
>
> We generated a completion for each prompt in the DeepScaleR dataset and recorded the number of times each token occurred. Below is a scatter plot showing cossim of each token’s embedding versus the occurrence count. Indeed, as you hypothesized, the lowest-cossim tokens are not frequent and do not meaningfully affect the overall performance. We thank you for this remark. We’ll work on finding the explanation for the earlier layers.
>
> By the way, 78% of tokens never got generated. 89% of tokens got generated <10 times. 96% of tokens got generated <100 times.
>
> https://8upload.com/image/dece67832fd980e3/cossim_vs_count.png

---

> ### Author Response · Authors · 2025-11-21
> **DiffSae**
>
> We decided to remove Section 8 as we agree that it requires further work.

---

> ### Author Response · Authors · 2025-11-21
> **Are they sink tokens?**
>
> We do not think there is a direct relation between them. Note that these tokens are the first generation tokens, not the first prompt tokens, which is the position that was previously shown to contain attention sink tokens.
>
> We’ve looked at how subsequent tokens attend to these special tokens, and there is no head on any layer that places a large attention weight on them. The largest fraction of tokens that look at the first generation token with a weight higher than 0.5 is around 0.6%.

---

> > ### Comment · Reviewer_TeJy · 2025-11-26
> >
> > Thank you for the clarifications.
> >
> > I remain my original score that the paper is not ready for publication at the moment, but believe that it can be strengthened substantially after revisions.

---

### Official Review · Reviewer_q2Fr · 2025-11-01

**Soundness:** 2
**Presentation:** 1
**Contribution:** 1
**Rating:** 2
**Confidence:** 3

**Summary:**

The paper studies steering vectors—small additive residual-stream interventions trained with an RL objective—and asks what they do inside math-reasoning LLMs. The authors train one vector per layer (Qwen2.5-Math-7B, LLaMA-3.1-8B-Instruct) on DeepScaleR, then probe with standard tools (logit lens, path patching, QK/OV lens, DiffSAE).

**Strengths:**

- The paper is carefully executed and empirically tidy.
- I think the paper does have some interesting findings. For example, the last layer behaves like first-token substitution (e.g., boosting “To”), and simply prefixing that token recovers roughly 75 % of the last-layer gain.

**Weaknesses:**

- The intellectual contribution feels incremental. The paper largely applies existing ingredients (e.g., bias-only adaptation/steering vectors trained with RL, standard logit-lens probes, path-patching, and QK/OV analysis) to known math-RL settings. The central insights (“early layers suppress non-English tokens,” “the last layer boosts a prompt-opening token,” “the penultimate layer acts via MLPs”) read as incremental characterizations of surface-level biases rather than new mechanisms or methods. I struggled to identify a new principle beyond careful confirmation of expected behaviors and left without a clear, portable mechanism or design rule that changes how we steer or train reasoning models.
- The DiffSAE results feel very preliminary and incomplete. The “feature associated with incorrectness” is intriguing but under-validated: features are described qualitatively (through a few examples), and the analysis focuses only on “the feature with the largest frequency difference.” There is no controlled group or ablation of any kind.
- The transfer experiments are mixed and weakly explained. Transfer appears to work well when the Qwen base model is the donor but fails for LLaMA, which the authors attribute to “different chat templates.” However, Qwen-Base and Qwen-Instruct also have different templates—so why does it work there? Moreover, since it does not work for LLaMA, the overall “vectors transfer” message feels overstated.

**Questions:**

Comments

- There are some formatting errors in the appendix (e.g., Appendix I) that need to be fixed, as well as in the references. Some text exceeds the page length.

- The paper also contains typos and a few incomplete sentences (e.g., line 174).

---

> ### Author Response · Authors · 2025-11-21
> **Scope**
>
> We scope our paper in the field of mechanistic interpretability, which aims to understand the internal computations or the changes to mechanisms, brought by some training procedure — in our case, RL for math reasoning. We did not intend to design a new method or improve the performance of training procedure, which is a matter for a separate paper with a different “spirit”.
>
> We contribute to the discussion (e.g., [1, 2]) about the sources of the high performance brought by RL reasoning, highlighting the factors that we found. To the best of our knowledge, the behaviours that we describe have not been mentioned in prior work and thus present novel findings worth sharing with the community.
>
> [1] Liu et al. “Understanding R1-Zero-Like Training: A Critical Perspective”
>
> [2] Venhoff et al. “Understanding Reasoning in Thinking Language Models via Steering Vectors”

---

> ### Author Response · Authors · 2025-11-21
> **DiffSAE**
>
> We decided to remove this section, as we agree that it requires a substantial improvement.

---

> ### Author Response · Authors · 2025-11-21
> **Transfer**
>
> All Qwen models in this experiment have the same chat template, specified in Appendix O.
>
> For a fair transferability test, we used R1-template for LLaMa3.1-8B-It, so now the templates are compatible. The table contains the transfer scores, showing non-trivial gains when exchanging steering vectors.
>
> |              | None        | Base         | Instruct     |
> | ------------ | ----------- | ------------ | ------------ |
> | **Base**     | 0.91 ± 0.11 | 9.18 ± 0.22  | 3.24 ± 0.27  |
> | **Instruct** | 3.33 ± 0.16 | 17.03 ± 0.34 | 21.76 ± 0.29 |

---

### Official Review · Reviewer_Pj2x · 2025-11-01

**Soundness:** 2
**Presentation:** 1
**Contribution:** 2
**Rating:** 2
**Confidence:** 4

**Summary:**

This paper analyzes steering vectors that are supposed to modify a models reasoning behavior and are trained via RL in a small vector that augments a frozen model and is inserted into the models' residual stream.

**Strengths:**

The related work is to my (limited) knowledge quite good and up-to-date.

**Weaknesses:**

Presentation-wise, the contributions on the first page are hard to understand. Overall, several parts of the paper should phrased more precisely and explain the line of thought more explicitly to make the paper easier to understand; In Section 7, I did not get the differences between the variations. The paper is also not really self-contained without the appendix. E.g., the takeaway of Section 8 is unclear from the section itself. My (personal) recommendation to the authors is to cut the paper to its main contributions and work out those in more detail. In its current form, it is trying to achieve to much but fails in providing detailed and generlizable insights about too little. I would strongly recommend the authors to always write out abbreviations at least at the first use (e.g., QK/QV, RLOO, even GRPO and MLP). This avoids ambiguities and reduces cognitive load.

The experimental study is done only on 3 models of two origins (LLama and Qwen). In order make results more generalizable, the experiments should be conducted with considerably models, specifically considering the partially non-aligning results. Regarding the impact of the approach, it should also be compared against other recent approaches mentioned in the related work to see what the effects are in comparison. Also different training datasets (for extracting the steering vectors) should be considered for the experiments.

Details:
I get no insights from Figure 8.
Figure 9: I would disencourage averaging results between benchmarks.
Examples are very hard to read on print-outs.

**Questions:**

Questions follow directly from the weaknesses.

---

> ### Author Response · Authors · 2025-11-21
> **Rephrasing Section 7**
>
> Let us rephrase the setup in Section 7:
>
> Steering the penultimate layer $s_{26}$ yields a larger accuracy gain than steering the last layer, and remains tractable to analyze because the modified activations traverse only one remaining layer. Here we identify which parts of that layer convert the steering signal into performance.
>
> For a given residual input $X$, the layer computes $Y = X + \text{MHA}(\text{LN}(X))$ and $Z = Y + \text{MLP}(\text{LN}(Y))$, where $\text{LN}$ denotes layer normalization. The Multi-Head Attention (MHA) mechanism consists of several attention heads, each defined as
>
> $H_i(U) = \text{Softmax}\!\left( \frac{U W_i^Q (U W_i^K)^\top}{\sqrt{d_k}} \right) U W_i^V,$
>
> where $W_i^Q$, $W_i^K$, and $W_i^V$ are the query, key, and value projection matrices for head $i$. The outputs of all heads are concatenated and linearly transformed using an output projection matrix $W^O$, forming the complete attention output:
>
> $\mathrm{MHA}(U) = [H_1(U); H_2(U); \ldots; H_h(U)] W^O.$
>
> The Multi-Layer Perceptron (MLP) sublayer applies a two-layer feedforward transformation of the form
>
> $\mathrm{MLP}(U) = f(U W_1 + b_1) W_2 + b_2,$
>
> where $f(\cdot)$ is a nonlinear activation function. To understand the contribution of each submodule, we insert or omit the steering vector $s_{l-1}$ at specific points within the layer and measure the resulting change in mean accuracy. If the change is significant, we mark the block as being important for processing the steering vector.
>
> We consider several setups. First, we denote the position for which the steering vector was pretrained as **All steering**: $X \leftarrow X + s_{26}$. **Skip-Attn**: $Y \leftarrow Y + s_{26}$ -- to assess the impact of the Multi-Head Attention (MHA) sublayer, we leave the input unchanged and apply the vector to the output of MHA. **Skip-Layer**: $Z \leftarrow Z + s_{26}$ -- similarly, we add the vector only to the output of the entire layer, evaluating its overall contribution compared to \textbf{All steering}, and isolating the effect of the MLP when compared to **Skip-Attn**. **Steer-Q/K/V-Proj**: $(U W_i^{Q/K/V}) \mapsto (U + s_{26}) W_i^{Q/K/V}$ — here, we examine the influence of each individual projection by steering only its input while keeping the others fixed.
>
> ---
>
> We have decided to remove Section 8 because we agree that its results are inconclusive. Also, thank you for pointing out other presentation flaws — we’ll fix them.

---

> > ### Author Response · Authors · 2025-11-21
> > **About models**
> >
> > We have an update regarding the vector transferability experiment: when using the same R1-style template (Appendix O) for both LLaMa models, exchanging the steering vectors now yields improvements over the respective base models. This resolves one of the previously noted inconsistencies between the families.
> >
> > |              | None        | Base         | Instruct     |
> > | ------------ | ----------- | ------------ | ------------ |
> > | **Base**     | 0.91 ± 0.11 | 9.18 ± 0.22  | 3.24 ± 0.27  |
> > | **Instruct** | 3.33 ± 0.16 | 17.03 ± 0.34 | 21.76 ± 0.29 |
> >
> > We selected these two since they are most widely used in the recent RLVR literature and using more models is resource-prohibitive. Still, we agree that extending the analysis to additional architectures is an important direction for future work.

---

### Official Review · Reviewer_sizY · 2025-11-02

**Soundness:** 2
**Presentation:** 2
**Contribution:** 2
**Rating:** 2
**Confidence:** 4

**Summary:**

The paper investigates the internal mechanisms by which RL enhances reasoning abilities of LLMs. The authors freeze the base models (Qwen2.5-Math-7B and Llama3.1-8B-Instruct), and train steering vectors added into the residual stream (independent of context) using a RL objective (RLOO). This goal is to isolate the RL-induced changes for mechanistic interpretation, instead of interpreting full fine-tuning. Note that the models trained by these different procedures would have different behaviors, so it's not clear that this would explain the behavior of RL-training. Nevertheless, the authors make interesting observations. Using mechinterp tools they find that (i) the last-layer vector operates primarily as a token-substitution bias boosting tokens like "To" or "Step"; (ii) the penultimate-layer vector operates mainly through the MLP pathway, largely bypassing attention mechanisms (though this was clearer in Qwen than Llama); and (iii) middle layers tend to de-emphasize non-English tokens. The authors show experiments on the transferability and composability of these vectors but get mixed results across the two model families.

**Strengths:**

- Using RL-trained steering vectors on a frozen base model is a new approach, and tractable for mechinterp.

- The analysis of the final two layers yields specific, concrete insights. The last-layer vector functions primarily as a first-token substitution (boosting "To"/"Step") is a cool finding, connecting well with "Thought Anchors" (Bogdan et al.).

- The circuit analysis (Section 7) showing that the penultimate vector in Qwen acts primarily through the MLP and bypasses attention mechanisms is interesting and perhaps useful to understand more deeply.

**Weaknesses:**

- The study aims to understand how standard RL fine-tuning changes internal computations. However, it analyzes a proxy: additive steering vectors trained with RL on a frozen model. The authors justify this by noting this approach can match the performance of full fine-tuning. However, equivalence of outcome does not imply equivalence of mechanism. Standard RL fine-tuning modifies weights distributively and can create complex new circuits. In contrast, steering vectors are highly constrained, additive interventions. So the observations may be tied to steering vectors and not general RL training.

- The paper only considers two model families (Qwen and Llama) and the findings are often inconsistent between the two (e.g., the penultimate layer mechanism was clear in Qwen but messy in Llama; composability was constructive in Qwen but often harmful in Llama). This makes it harder to see if the results are model specific or actually about the training procedure.

- Some interpretations are questionable or spurious. The finding that middle layers de-emphasize non-English tokens is likely an artifact of the training data (English math problems) rather than a fundamental mechanism of reasoning. The interpretation of the DiffSAE features is also speculative and weakly supported by the ambiguous evidence they provide in Figure 7.

- The paper is generally clearly written but currently reads like a collection of isolated experiments, and could benefit from better motivation.

**Questions:**

Beyond the concerns in the weaknesses, here are some more questions:

- In Figure 1, 23, 24th layer has low mean accuracy, and it is explained that the LayerNorm in 23rd layer limits the steering performance? Why for other layers, the layer norm would not affect the performance?

- In Figure 4, “To” and “Step” are boosted which is interesting and might be related to reasoning. However, a more in-depth analysis is lacking (Between Line 293-300), would other “thought anchors” (Bogdan et al.) be also boosted ?

- In Figure 5, it is shown that for the penultimate layer, steering effect is mainly on MLP rather on attention. Is this also true for all layers?

- Figure 7 seems not evident enough to conclude they are mostly related to incorrect generations. For example, in the 10-th layer, there are also 50% of the correct ones are activated.

---

> ### Author Response · Authors · 2025-11-21
> **About proxy**
>
> While we indeed study a proxy, we see this setup as informative in two ways. First, the steering vectors are easier to interpret, since in our parameterization all RL-induced changes are isolated into a few trained parameters. Second, in our experiments they implement a simple mechanism that is sufficient to recover most of the reasoning gains. As you note, this mechanism is reminiscent of “thought anchors” or [1] found in fully trained models, which suggests it is relevant beyond our specific proxy. We therefore believe it is important for future work on interpreting RL training to treat such simple mechanisms as baseline explanations before positing more complex ones.
>
> [1] Ji et al. “The First Few Tokens Are All You Need for Fine-Tuning Reasoning Models”

---

> ### Author Response · Authors · 2025-11-21
> **About models**
>
> We have an update regarding the vector transferability experiment: when using the same R1-style template (Appendix O) for both LLaMa models, exchanging the steering vectors now yields improvements over the respective base models. This resolves one of the previously noted inconsistencies between the families. Beyond this, we expect some observations to differ across Qwen and LLaMa, since in our experiments the two families reach different performance ceilings under RLVR. We interpret these discrepancies as reflecting different realizations of the RL-induced strategy that depend on architectural and pretraining properties of the base models.
>
> |              | None        | Base         | Instruct     |
> | ------------ | ----------- | ------------ | ------------ |
> | **Base**     | 0.91 ± 0.11 | 9.18 ± 0.22  | 3.24 ± 0.27  |
> | **Instruct** | 3.33 ± 0.16 | 17.03 ± 0.34 | 21.76 ± 0.29 |
>
> We selected these two since they are most widely used in the recent RLVR literature and using more models is resource-prohibitive. Still, we agree that extending the analysis to additional architectures is an important direction for future work.

---

> ### Author Response · Authors · 2025-11-21
> **Interpretations**
>
> Indeed, this result is dataset-specific, but we still see it informative — if such a simple strategy explains a portion of performance gap, we take it as a valid explanation. Moreover, many papers from the RLVR literature are developed with datasets similar to DeepScaleR and mech-interp works conduct their analysis on a single dataset (like [1] and [2], also similar to DeepScaleR). We consider it is important to highlight that part of the performance is explained by dataset-specific biases and heuristics in addition to more complex explanations.
>
> Regarding DiffSAE, we agree with your criticism and we decided to remove this section and to focus on the other better-supported findings.
>
> [1] Bogdan et al. “Thought Anchors: Which LLM Reasoning Steps Matter?”
>
> [2] Venhoff et al. “Understanding Reasoning in Thinking Language Models via Steering Vectors

---

> ### Author Response · Authors · 2025-11-21
>
> **Layers 23-24**
>
> We do not yet have a definitive explanation. Note that we are not claiming that LayerNorm itself blocks performance; rather, our results suggest that the degradation at layers 23–24 arises from the LayerNorm in layer 25.
>
> **Is it true for all layers?**
>
> Unfortunately, we have not run this analysis for all layers, as it would require a large sweep over positions and each run is computationally expensive.
>
> **About Figure 7**
>
> We have decided to remove Section 8 because we agree that its results are inconclusive.

---

> ### Author Response · Authors · 2025-11-21
> **Other "thought anchors"**
>
> To address this question, we trained another Qwen2.5-Math-7B model in which all prompts were appended with the token “To”, so that the last-layer steering vector could no longer improve performance simply by boosting “To”. This forces the vector to learn other preferences.
>
> Figure 1 shows the top-15 token substitutions induced by the last-layer steering vector, sorted by the maximum probability difference; the number of occurrences of each substitution is given after the hash sign. Many high-impact edits replace tokens where punctuation is merged with two newline symbols by just the punctuation mark. In other cases, there is an exchange of words, seemingly shortening generation and making it “straight to the point”. Figures 2 and 3 shows changes for specific tokens on example texts.
>
> Figure 1: https://8upload.com/image/66ed0a11089e0d9d/slash-ns.png
>
> Figure 2: https://8upload.com/image/0ff098ce946bf7a5/other-thought-anchors-1.png
>
> Figure 3: https://8upload.com/image/0eb3f967bd07cb45/other-thought-anchors-2.png

---

### Meta-Review · Area_Chair_C8Ma · 2026-01-05

**Summary:**

The reviewers were broadly aligned that the paper contains several interesting empirical observations about RL-trained steering vectors, especially the concrete analyses of late-layer effects. However, the consensus was that the current submission does not yet meet the bar for ICLR due to (i) central methodological limitations; (ii) limited scope and generality; (iii) several claims/sections originally being weakly supported or overstated, and (iv) presentation and narrative issues

**Reviewer Concerns:**

Reviewer sizY
Addressed in rebuttal / revision: DiffSAE concerns; Transfer inconsistency; “Thought anchors” depth; LayerNorm question.
Still outstanding: Proxy-to-RL-finetuning gap remains the central unresolved issue; Generality / consistency across model families; Narrative cohesion

Reviewer Pj2x
Addressed in rebuttal / revision: Clarity of Section 7; DiffSAE / unclear Section 8. Still outstanding: Presentation and organization; Limited breadth of evidence; Over-aggregation / figure readability

Reviewer q2Fr
Addressed in rebuttal / revision: DiffSAE concerns; Transfer explanation; Scope clarification. Still outstanding: Incremental intellectual contribution; “Vectors transfer” and related broad claims;

Reviewer TeJy
Addressed in rebuttal / revision: Non-English token downweighting concern; Sink-token question; DiffSAE concerns. Still outstanding: Insufficient evidence for some claims; Lack of actionable/portable takeaways; Overall readiness for publication

**Reviewer Scores:**

Reviewer sizY: stays at 2.
Reviewer Pj2x: stays at 2.
Reviewer q2Fr: stays at 2.
Reviewer TeJy: stays at 2 (reviewer explicitly maintained score after rebuttal).

---

### Decision · Program_Chairs · 2026-01-26

Reject